# Ensemble hindcasting of wind and wave conditions with WRF and WAVEWATCH III® driven by ERA5

Robert Daniel Osinski[1] and Hagen Radtke[1]

[1]Leibniz Institute for Baltic Sea Research Warnemünde, Seestrasse 15, 18119 Rostock, Germany

**Correspondence:** Robert Daniel Osinski (robert.osinski@io-warnemuende.de)

**Abstract.** When hindcasting wave fields of storm events with state-of-the-art wave models, the quality of the results strongly depends on the meteorological forcing dataset. The wave model will inherit the uncertainty of the atmospheric data, and additional discretization errors will be introduced due to a limited spatial and temporal resolution of the forcing data. In this study, we apply an atmospheric downscaling to (i) add regional details to the wind field, (ii) increase the temporal resolution of the wind fields, (iii) provide a more detailed representation of transient events such as storms, and (iv) generate ensembles with perturbed atmospheric conditions which allow for a flow dependent and spatiotemporally variable uncertainty estimation. We test different strategies to generate an ensemble hindcast of a relatively strong storm event in February 2002 in the Baltic Sea. The Weather Research and Forecasting (WRF) model used for this purpose is driven by the ECMWF ERA5 reanalysis, and wind fields are passed to the third-generation wave model WAVEWATCH III®. A combination of initial conditions from the ERA5 ensemble of data assimilations and stochastic pertubations during runtime is identified as the most promising strategy. The final aim of the ensemble approach is to quantify the hindcast error, but this approach can also be used to generate alternative representations of historical extreme events to sample the recent climate and to increase the sample size for statistical studies, such as for civil engineering applications for coastal protection studies.

## 1 Introduction

The Lorenz attractor (Lorenz, 1963) is often used as an example to motivate ensemble forecasts. It explains a chaotic system behaviour, which is very sensitive to slight differences in the initial conditions and is described by a system of differential equations. In operational weather prediction, ensemble forecasts are a common tool to quantify the forecast uncertainty by producing a set of alternative realizations. Initial conditions are estimated by data assimilation combining observations with a background field, which is normally a previous model run. The sparse spatiotemporal observational coverage leads to uncertainties in the initial conditions, which are growing over the integration time. A second type of uncertainty comes from the model parametrizations. These are used to take processes into account, which cannot be resolved by the dynamical core of the model, e.g. subgrid-scale processes like turbulence or convection, or processes which can be described physically but are computationally too expensive to explicitly take them into account (e.g. utilization of a 1-moment instead of a 2-moment micro physics scheme).

In principle, three methods exist to generate an ensemble forecast, the latter two of which are these are tested in this communication to estimate the uncertainty of a hindcast. The first possibility is the combination of forecasts from different models (e.g. Hagedorn et al., 2005) or using the same model with different types of model physics (e.g. Ricchi et al., 2019). This multi-model/physics approach has the disadvantage that the ensemble size is limited to the number of available models/physics

packages. Also, the forecast skill over a specific region and a specific variable might differ between the different models, what has to be taken into account in the interpretation. A second approach is the combination of forecast runs from the same model for the same time instance, but started at different initialization times, called lagged-average forecast (LAF) ensemble (Hoffman and Kalnay, 1983). A limitation here is also the number of forecasts covering the same time instance and the fact that a newer forecast can be expected to have in average a better forecast skill than a forecast at long lead times. The third method is

the utilization of a single model and applying pertubations to the initial conditions and/or to the model physics.

Such an approach is used operationally at the European Centre for Medium-Range Weather Forecasts (ECMWF) since 1992. Initial conditions are perturbed by singular vectors (Buizza, 1998) or by a combination of Ensemble data assimilation (Buizza et al., 2008) with singular vectors, or by breeding vectors (Toth and Kalnay, 1997) like in case of the National Centers for Environmental Prediction (NCEP). Stochastic perturbations like Stochastically Perturbed Parametrization Tendencies (SPPT)

(Buizza et al., 1999), Stochastically Perturbed Parametrizations (SPP) (Ollinaho et al., 2017) and Stochastic Kinetic Energy Backscatter (SKEB) (Shutts, 2005) are used to perturb the model physics (Leutbecher et al., 2016, 2017). SPPT perturbs the model parametrizations by applying a multiplicative noise and SKEB simulates the upscale transfer of kinetic energy from smaller to larger scales. Besides the application of SPPT in the global ECMWF medium-range ensemble system, stochastic perturbations are also used in local area models (e.g. Bouttier et al., 2012) and in ocean models like in NEMO (e.g. Brankart

et al., 2015).

In a well constructed ensemble, the ensemble spread reflects the average forecast error. Stochastic perturbations need some time until a reasonable spread develops. Ensemble data assimilation (EDA) gives different estimations of the initial state representing its uncertainty. A forecast started from the different members develops the desired ensemble spread faster.

ERA5 (Copernicus Climate Change Service (C3S), 2017) is the newest global reanalysis from ECMWF. The resolution is

relatively high with about 31 km resolution for the atmospheric variables, but depending on the application, it can be still too coarse. From ERA5, in contrast to previous reanalyses, an uncertainty measure based on an ensemble of data assimilation is available.

The Weather Research and Forecasting (WRF) (Skamarock et al., 2019) model is widely used in research as well as in operational weather forecasting and includes implementations of the mentioned stochastic perturbation schemes. The motivation

of driving WRF with this new dataset is to be able to produce hindcasts of atmospheric conditions in different spatiotemporal resolutions including a measure of uncertainty based on ensemble techniques. This allows for example to study the effect of the model resolution on effects like up- and downwelling in coastal regions.

Some regional reanalysis (ensemble) datasets are already freely available. Such regional reanalyses are produced, for example, in the framework of the project "Uncertainties in Ensembles of Regional ReAnalysis" (UERRA)[1]. At the moment, the

---

[1]http://www.uerra.eu/

ensemble datasets in this project are limited in their temporal coverage or spatial resolution. It can be advantageous to be able to produce hindcasts of events whose spatiotemporal resolution is adapted to the requirements defined by a research objective. It has to be mentioned that the quality of a freely running hindcast can be expected to be inferior to such a re-/analysis product containing state-of-the art data assimilation techniques. Another database from which ensemble forecasts of local area models

are available is from the Tigge-LAM archive[2] (Swinbank et al., 2016). The available forecast models cover also only short periods and they are operational, meaning that the datasets are not homogeneous, because the model version can change during time.

The Baltic Sea, which is a marginal sea in the north-east of Europe, is taken as an example for the application of the demonstrated procedure to produce ensemble hindcasts of wind and wave conditions by driving the WAVEWATCH III® wave

model with wind data produced with the WRF ensemble model. Observed wave heights in this region do not exceed 8.2 m (Björkqvist et al., 2017) and waves are dominated by the wind sea (Broman et al., 2006; Soomere et al., 2012). More detailed information about the Baltic Sea wave climate for specific subregions is provided, for example, by Björkqvist et al. (2017), Soomere (2005), Soomere et al. (2008), Tuomi et al. (2011) and Tuomi et al. (2014). As ERA5 is a global reanalyis, the demonstrated procedure is also applicable in other regions.

The idea behind this study is to generate an ensemble hindcast on event basis in a comparable way to operational weather forecasts by driving WRF with ERA5 including the initial conditions from the ERA5 EDA with stochastic perturbations (SKEB and SPPT). Other ensemble generation techniques are tested for comparison. The atmospheric data from a hindcast or forecast are discrete in time and space. This limits the accuracy and affects the outcome if driving another model like an ocean or wave model. This uncertainty is investigated by driving the wave model with different spatiotemporal resolutions.

## 2   Data and models

### 2.1   Data

#### 2.1.1   ERA5

ERA5[3] (C3S, 2017) is the follow-up ECMWF reanalysis of ERA-Interim produced with the Integrated Forecasting System (IFS) cycle 41R2[4], operationally at ECMWF in March 2016. It is provided under the Copernicus licence[5] allowing also com-

mercial applications. Hourly reanalysis in about 31 km[6] ($\sim$0.28°) horizontal resolution and 137 vertical model levels are available from 1979 (eventually 1950) and the dataset is getting prolongated into the future with a delay of about three months. A state-of-the-art data assimilation technique is used (4D-Var). In addition to the reanalysis, on three hourly basis, ten members

---

[2]https://apps.ecmwf.int/datasets/data/tigge-lam/expver=prod/type=pf/

[3]https://confluence.ecmwf.int/display/CKB/ERA5+data+documentation

[4]https://www.ecmwf.int/en/forecasts/documentation-and-support/changes-ecmwf-model/ifs-documentation

[5]http://apps.ecmwf.int/datasets/licences/copernicus/

[6]Grid cells in the Baltic Sea region have quite large aspect ratios, the length of their sides in N-S direction can be roughly twice as long as in the E-W directions.

of an ensemble of data assimilation (EDA) are provided as an uncertainty measure with half of the resolution of the reanalysis. The reanalysis data of surface fields and the 137 model levels were extracted on hourly basis interpolated onto a slightly higher 0.25° resolution grid for the period 21 February 2002 until 24 February 2002 as recommended by ECMWF. ERA5 data from the ensemble of data assimilation were also interpolated bilinearly onto the same 0.25° regular longitude-latitude grid. ERA5 also includes fields from the ECWAM wave model (ECMWF, 2016) in 0.36° and in 1° spatial resolution from the ensemble of data assimilation. ERA5 is used for the initial and lateral boundary conditions for the atmospheric hindcasts with the WRF model. Lateral boundary conditions for the Baltic Sea WAVEWATCH III® setup originate from a setup for the North Sea. This coarser model is driven by ERA5 winds. ERA5 reanalysis and EDA wind and wave data are used for comparison of the hindcasts produced with WRF and WAVEWATCH III®.

### 2.1.2   UERRA/Harmonie-v1

The UERRA/Harmonie-v1 dataset (Ridal et al., 2017) contains analyses at 00, 06, 12 and 18 UTC as well as hourly forecasts for +1h until +6h and thereafter three-hourly until thirty hours. The Harmonie model is used for the production of this dataset in about 11 km horizontal resolution and 3D-Var data assimilation is used with conventional observations (synoptic stations, ships, drifting buoys, aircraft observations and radio soundings). Large scales from ERA40 and ERA-Interim are introduced into the data assimilation by large scale mixing. The available period extends back until 1961. To create an hourly dataset, the analysis fields were combined with the forecast lead times +1h to +5h, retrieved from ECMWF[7]. Wind data were interpolated bilinearly onto the regular wave model grid for the Baltic Sea described in the next section. This dataset was mainly used to produce a restart file for the wave model runs and for calibration/validation of the wave model.

### 2.2   Models

### 2.2.1   Atmospheric Weather Research and Forecasting model (WRF)

The Weather Research and Forecasting model WRF v4.0.3 model[8] in the Advanced Research WRF (ARW) version (Skamarock et al., 2019) is applied here. It is used in non-hydrostatic mode in 0.126° horizontal resolution and the model output interval is 15 minutes. To investigate the dependence of the solution of the wave model on the spatial and on the temporal resolution of the wind data, runs in 0.252° and 0.063° were produced as well as output at a temporal resolution of 5 minutes. In this way, a factor of about 4.5 between the highest WRF resolution and the driving ERA5 fields is given, and the same factor between the ERA5 EDA fields and the WRF ensemble runs. The domain is slightly larger than the Baltic Sea for all runs. For the model configuration, the CONUS physics suite (Wang et al., 2019) is used. This is a combination of model physics adapted for the Continental United States of America. As it is well tested, this physics setup is taken as it is, and we assume that it should be reasonable for other regions in the mid-latitudes. The 89 vertical Eta layers used in this WRF setup, a specific vertical coordinate system in atmospheric models, are adapted to be comparable to layer 2 to 90 of the IFS[9] until 50 hPa. As initial

---

[7]https://apps.ecmwf.int/datasets/data/uerra

[8]https://github.com/wrf-model/WRF/releases

[9]https://www.ecmwf.int/en/forecasts/documentation-and-support/137-model-levels

conditions come from a different and coarser model, it needs some time until fine structures develop. Methods for spin-up reduction like Digitial Filter Initialization (Peckham et al., 2016) have not been tested. Instead, the WRF output is only used twelve hours after initialization to drive the wave model. Neither data assimilation nor observation nudging is used. Hindcasts are produced in this study in a comparable way like a forecast, downscaled from a global forecast model. For this reason, the

results of this study are valid for both hindcasts and such forecasts. The WRF Pre-Processing System (WPS) in version 4.0.3 is used to prepare the input data for the model together with the WPS V4 Geographical Static Data[10].

### 2.2.2    Wave model WAVEWATCH III®

WAVEWATCH III v6.07® [11] (Tolman, 1991; The WAVEWATCH III® Development Group (WW3DG), 2019) is used in this study for the Baltic Sea. It is a state-of-the art third generation wave model, which is also used as an operational wave forecast

model. A one-way nesting approach is applied, see Figure 1: A setup with 0.1° resolution covering the North Sea and a small part of the eastern Atlantic ocean is used to produce boundary conditions for the Baltic Sea setup at the border with the North Sea. This is not really necessary for the central and northern regions of the Baltic Sea, as very little wave energy passes the Danish straits. To avoid showing unrealistic values in a part of the domain, the nesting procedure was nevertheless applied. The GEBCO_2014 Grid in version 20150318[12] is used as bathymetry. The Baltic Sea setup has a resolution of one nautical mile with

149.282 sea grid points and the bathymetry is based on the work of Seifert et al. (2001). UERRA/Harmonie-v1 was used for calibration and validation of the setup against one month of data from buoys available from the Copernicus Marine environment monitoring service[13] (CMEMS) with the previous WAVEWATCH III v5.16 version. A calibration and validation with the WRF forcing was not yet possible because of the short period that has been hindcasted until now. Nevertheless, the wave model shows a satisfactory performance with the WRF forcing. Detailed information about the calibration and validation procedure of the

wave model can be found in the supplemental material. 24 directions starting at 7.5° with a 15° direction increment and 30 frequencies starting at 0.03453 Hz geometrically distributed with a step of 1.1 are used for the discretization of the energy spectrum. This is comparable to the settings for the wave model in ERA5. Soomere (2005) proposes a finer resolution of the energy spectrum. This finer resolution was tested and the result is demonstrated in the supplemental material. A clear impact on the extreme wave heights is visible, but it prolongates significantly the computing time. For our specific application, the ERA5

discretization is a good compromise between computational effort and model performance. The physics packages are defined before compiling the model by a so-called switch file. The switch file Ifremer1, provided with the model code, is applied in this study. This includes wind input and dissipation after Ardhuin et al. (2010) and the SHOWEX bottom friction scheme (Ardhuin et al., 2003). A sediment map based on the European Marine Observation and Data Network EMODnet[14] data was used for applying non-homogeneous bottom friction. The model runs were produced between 22 February 2002 00 UTC and

---

[10]http://www2.mmm.ucar.edu/wrf/src/wps_files/geog_high_res_mandatory.tar.gz

[11]https://github.com/NOAA-EMC/WW3

[12]http://www.gebco.net

[13]http://marine.copernicus.eu/services-portfolio/access-to-products/?option=com_csw&view=details&product_id=INSITU_BAL_NRT_
OBSERVATIONS_013_032

[14]http://www.emodnet-geology.eu/

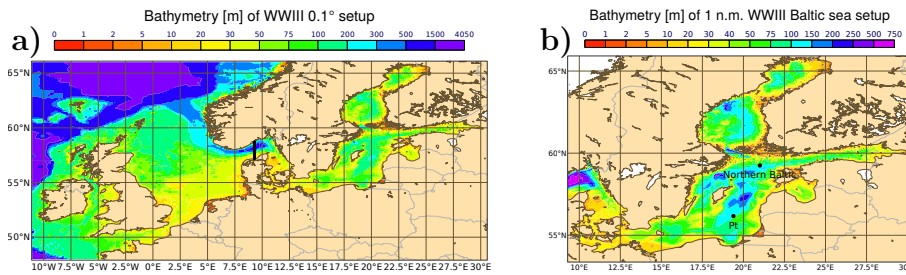

**Figure 1.** Bathymetries [m] and domains of a) 0.1° and b) 1 nautical mile WAVEWATCH III® setups; in black in the left panel a) grid cells for the nesting of the Baltic Sea model are shown. The black point "Pt" in the right panel b) shows the location of the time series in Figure 2 and the black point "Northern Baltic" the location of the time series in Figure 6.

24 February 2002 00 UTC. WAVEWATCH III® was started from initial conditions from a previous run conducted for 21 days driven with UERRA/Harmonie-v1. The sea ice area fraction is taken from ERA5. In the atmospheric model, the stochastic perturbations of the model physics contribute significantly to the ensemble spread. Wave models include different source terms (e.g. wave generation, dissipation, bottom-friction, and so on), which are partly simplified to make the model computationally

more efficient or are described empirically (Farina, 2002; Yildirim and Karniadakis, 2015). Nevertheless, the wave model ensemble approach here is based solely on the ensemble of the atmospheric forcing data and includes no perturbations of the source terms.

## 3   Ensemble hindcasts

### 3.1   Wind fields

Six different approaches to generate an ensemble hindcast are presented in this section, see Table 1. The first approach is to generate an LAF ensemble. This is done by initialising the WRF model at different times on 21 February 2002 at every hour between 08 and 16 UTC, which results in 9 runs covering the period from 22 to 24 February 2002. The second approach is based on the domain shifting presented by Pardowitz et al. (2016). The ERA5 reanalysis is for this purpose shifted by one grid cell (0.25°) in each direction horizontally producing 8 perturbed ensemble members. For the third approach, WRF is initialised

from the ERA5 fields from the ensemble of data assimilation. These fields have a coarser resolution, but they are used in this study as the ERA5 reanalysis in 0.25°. This has the disadvantage that finer scales are not represented, but this is comparable to a downscaling from a global ensemble model, except that the reanalysis is used here as lateral boundary condition. As an alternative to keep the finer scales, it was tested to add perturbations to the initial fields, calculated by the difference between the ERA5 EDA members and the EDA ensemble mean, once with positive and negative sign to the ERA5 HRES reanalysis.

We didn't find an improvement against the direct application of the ERA5 EDA fields. SKEB and SPPT are used for the fourth approach and the fifth combines approach three and four. For approach six, the same setup is used as in approach five, but

**Table 1.** Methods tested for the generation of an ensemble hindcast with WRF

| Method | Procedure | No. of members |
|--------|-----------|----------------|
| 1 | LAF approach, WRF initialised at different times (21 February 2002 between 08 and 16 UTC) | 9 |
| 2 | Domain shifting approach, ERA5 shifted horizontally by one grid cell | 8 |
| 3 | ERA5 EDA fields used for initial conditions | 10 |
| 4 | Stochastic perturbations (SKEB and SPPT) together with random perturbations of LBC's | 10 |
| 5 | As approach 4, but initialised from ERA5 EDA as in approach 3 | 10 |
| 6 | As approach 5, but additional runs started at three hours earlier and later | 30 |

as the ERA5 EDA fields are available every three hours, runs three hours earlier and later are additionally used as in an LAF approach. This leads to a thirty member ensemble.

In an ensemble system, it is important that the ensemble spread reflects the uncertainty. If the spread is too narrow, the system is underdispersive meaning that the forecast is overconfident and vice versa for an overdispersive/underconfident forecast. One tool for quantifying the quality of the ensemble spread is, for example, the Talagrand (rank) diagram (Hamill, 2001), and there are other quality measures like for example reliability, resolution, accuracy or sharpness, which are important for a good ensemble system (Murphy and Winkler, 1992). To be able to use the traditional ensemble verification methods (Jolliffe and Stephenson, 2003), long time series are needed, which could not be produced in this study. For this reason, an absolute statement which of the tested approaches performs best can not be given based on only one single hindcasted event. The different approaches are compared against the ERA5 reanalysis and the ERA5 members from the ensemble of data assimilation. As a larger variability can be expected in the higher-resolution model, it can be assumed that it increases also the uncertainty, what should be reflected by a larger spread than found in the much coarser data from the ERA5 ensemble of data assimilation.

A good agreement at a specific location between the ERA5 reanalysis and the WRF runs is visible during the first twenty hours in Figure 2. The wind speed maximum is higher than in ERA5. For comparison, the closest grid cell of the UERRA/Harmonie-v1 data is plotted and also shows higher values than ERA5. From ERA5, also the wind speed from the closest grid cell of the 0.25° grid is plotted. The initial conditions were prepared with the WRF preprocessing system and can have slightly different values from taking simply the closest grid cell. The resolution of the WRF runs is closer to the one from UERRA/Harmonie-v1, and a stronger variability and also higher extremes can be expected due to the difference to the ERA5 resolution. WRF adds additional information from the finer scales and resolves the orography, coastlines and islands better.

All ensemble techniques lead to deviations from the unperturbed run. The LAF ensemble shows a very small spread. In fact, this is good, because it means that irrespective of the starting time of the WRF model being shifted by a few hours, the outcome is comparable. The first three approaches show a lower ensemble spread than the ensembles which include stochastic perturbations. Compared to the ERA5 EDA members, it demonstrates that these ensembles are underdispersive. The uncertainty is underestimated by applying these approaches. During the first hours of the ensemble with only stochastic perturbations

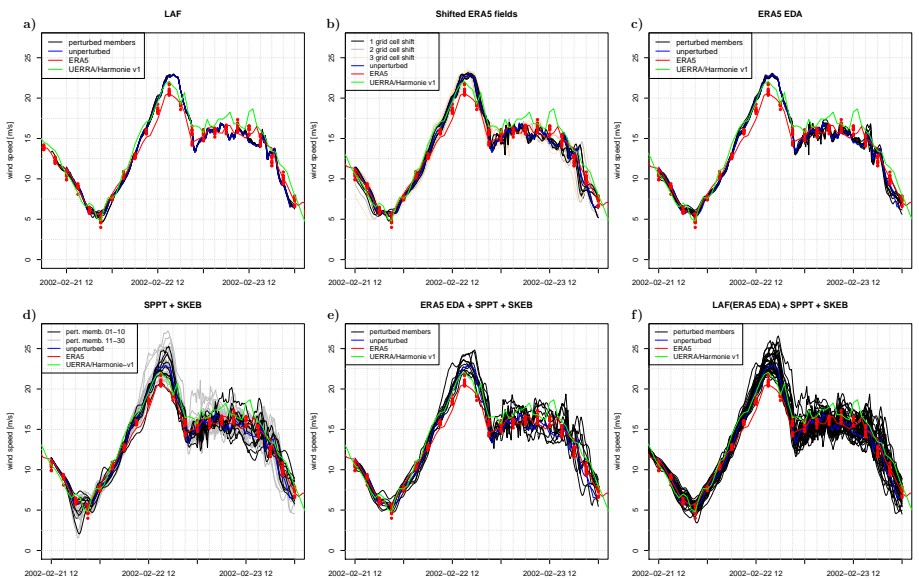

**Figure 2.** Time series demonstrating the simulation results of the different ensemble generation strategies at one specific location: a) lagged-average forecast (LAF) ensemble (Hoffman and Kalnay, 1983), b) domain shifting (Pardowitz et al., 2016), c) WRF runs started from ten ERA5 4D-EnVAR members with HRES LBC's, d) stochastic perturbations (SPPT and SKEB) (Buizza et al., 1999; Shutts, 2005), e) ERA5 4D-EnVAR as starting conditions plus stochastic perturbations, and d) LAF started from ERA5 4D-EnVAR at 09, 12 and 15 UTC plus stochastic perturbations; results shown at 19.39°E, 56.17°N

(ensemble approach 4), all members are identical, as it needs some time until the perturbations introduce spread. Starting from the ERA5 ensemble of data assimilation (ensemble approaches 3, 5, and 6), spread is visible from initialization on. Even with the coarser resolution of these fields, its application seems to be working, but additionally stochastic perturbations are necessary to produce a larger spread. The WRF ensemble started from ERA5 EDA fields at 09, 12 and 15 UTC also represents

5  the uncertainty at 21 February 2002 at 21 UTC, where the lowest values in the ERA5 EDA members (Fig. 2) in the shown period can be found. With only stochastic perturbations, such low values are also visible, but a few hours too early. For the last simulation day, the spread of the combined ERA5 EDA and stochastic perturbations approach is very large, but it could not be tested if it is overdispersive.

Spatially (Fig. 3), the spread in the WRF ensemble started from ERA5 EDA is very small. A much larger spatial variability

10  appears by applying stochastic perturbations. Especially strong wind is present in some members over the northern part of the Baltic Sea. The LAF approach also shows very little spread spatially over the entire domain. Domain shifting also did not produce as strong variability as applying stochastic perturbations. The combination of ERA5 EDA and stochastic perturbations produces members which show a strong variability in the central Baltic Sea (Fig. 4).

The LAF approach also shows very little spread spatially over the entire domain, and domain shifting also did not produce

15  as strong variability as applying stochastic perturbations (Fig. 3).

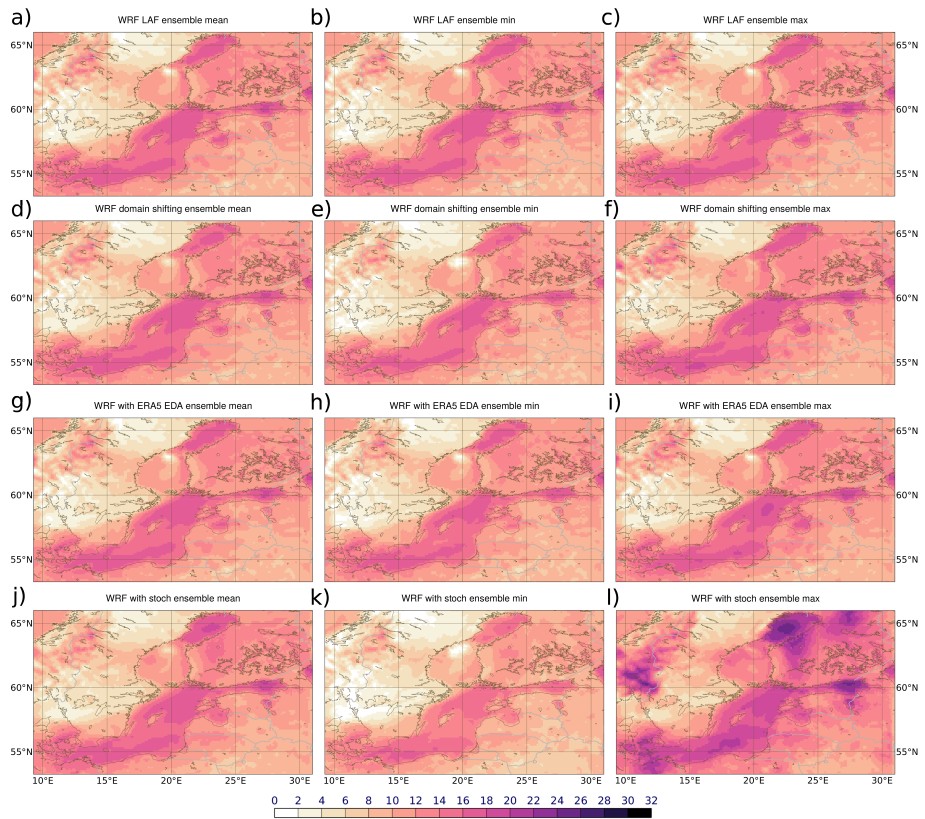

**Figure 3.** Ensemble mean a,d,g,j), minimum b,e,h,k) and maximum c,f,i,l) wind speed [m/s] of WRF ensemble based on LAF approach a-c), domain shifting d-f), initial conditions from the ERA5 EDA g-i) and on j-l) stochastic perturbations. All initialised at 21 February 2002 12 UTC. Shown 23 February 2002 09 UTC.

A strong variability in the Northern as well as in the Central Baltic Sea is present by initialising WRF at 09, 12 and 15 UTC from ERA5 EDA fields with stochastic perturbations. Ten members are a small number to sample the uncertainty.

Comparing a ten with a thirty member ensemble is not really a fair comparison, as a too small ensemble size leads to an undersampling of the uncertainty. Ensemble approach 6 shows in the ensemble maximum high values in the central as well as in the northern part of the Baltic Sea. Figure 2 shows also the WRF ensemble with only stochastic perturbations and thirty members. The spread is in this case larger, but still inferior to the thirty member approach number 6 with ERA5 EDA as initial conditions and stochastic perturbations. Also the region in the central Baltic Sea gains spread by adding additional members, but contains lower spread than in approach 6 shown in Figure 5. This demonstrates that ten members might be still not sufficient to sample the entire range of uncertainty, and that the combined application of model and initial perturbations is beneficial to create a larger spread.

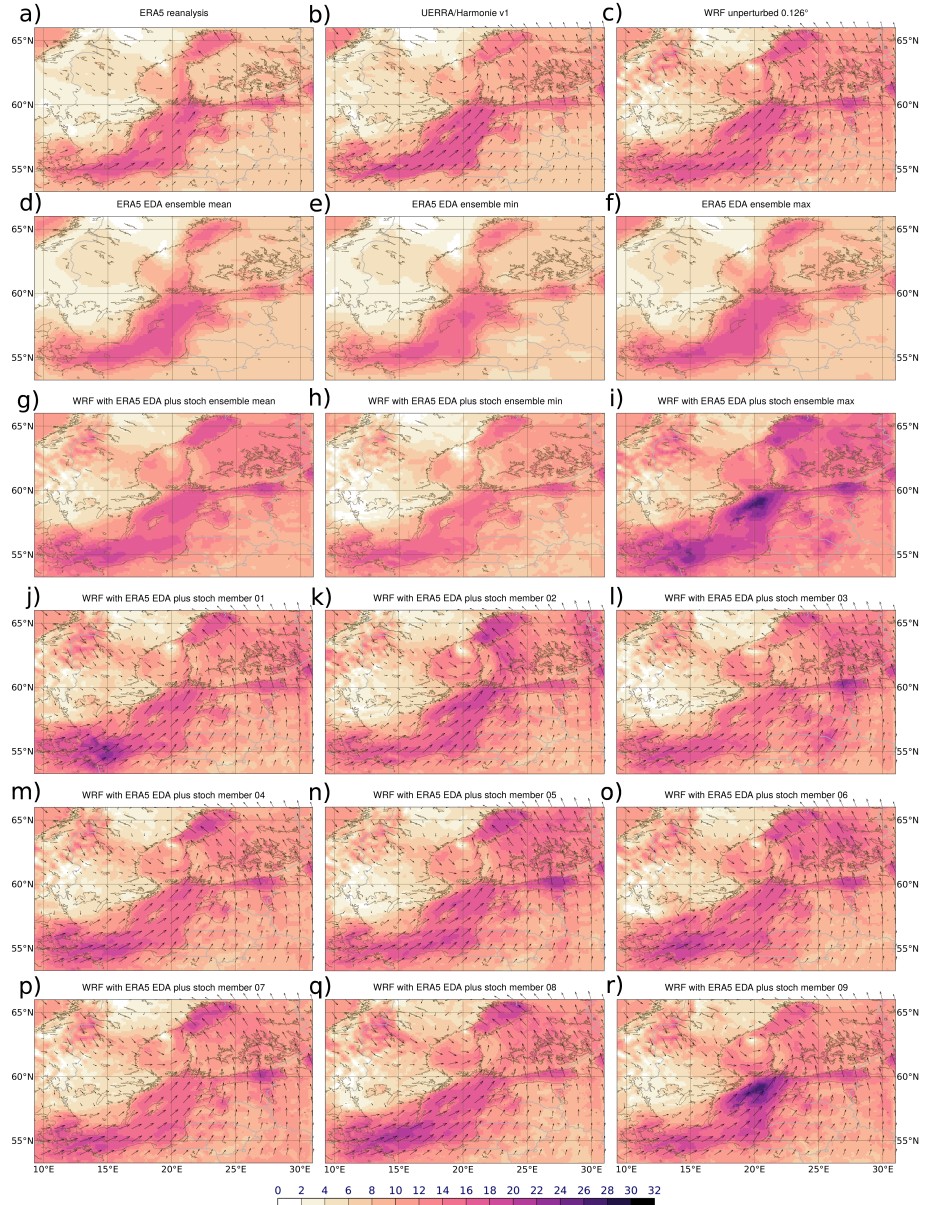

**Figure 4.** Postage Stamps: WRF ensemble approach number five generated by starting the ten members from ten ERA5 EDA members at 21 February 2002 12 UTC plus stochastic perturbations SPPT and SKEB. Shown 23 February 2002 09 UTC. a) The ERA5 reanalysis, ERA5 EDA ensemble d) mean, e) minimum and f) maximum, c) WRF unperturbed, WRF g) ensemble mean, h) minimum and i) maximum, b) UERRA/Harmonie-v1, and nine j-r) perturbed WRF members are shown. Wind speed [m/s] and direction as arrows.

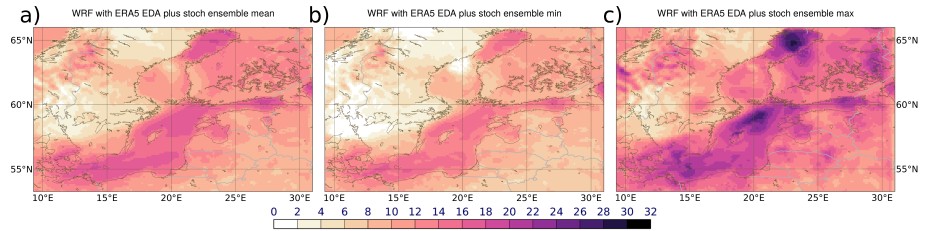

**Figure 5.** Ensemble mean a), minimum b) and maximum c) wind speed [m/s] from WRF ensemble approach number 6 generated by starting three times ten members from ERA5 EDA members at 21 February 2002 09/12/15 UTC plus stochastic perturbations SPPT and SKEB. Shown 23 February 2002 09 UTC.

## 3.2 Wave fields

The LAF and domain shifting approaches were not used to drive the wave model, because they show a relatively small spread. Figure 6 shows a time series at a location in the central Baltic Sea (see Figure 1). The comparison with the closest grid cell from ERA5 shows a good agreement in the temporal evolution of growth and a comparable trend in the decay of the significant wave height, but the maximum peak is about one metre lower in ERA5. ERA5 also shows the second peak only very weakly and some hours later during the middle of the second simulation day. WAVEWATCH III® in this study has a much higher resolution with 1 n.m. compared to the 0.36° ECWAM model of the ERA5 reanalysis and the WRF wind forcing is spatially (0.126° vs. 0.28°) and temporally (15' vs. 60') of higher resolution. This can explain locally much higher values and a stronger variability. Especially the maximum of the (wind speed and the) significant wave height varies strongly between the different ensemble realizations. Differences in the wave fields of the ensemble members can be due to a different dynamical evolution of the storm or due to different tracks in the atmospheric model members (compare Osinski et al., 2016). Already a slight change in the track of the storm can provoke large differences in the maximum if looking at a specific location in such a high resolution. With 0.36° resolution in ERA5, a slight change in the track can be assumed to not lead to such strong differences.

Figure 7 shows the different wave model members driven by WRF with ERA5 EDA initial conditions and stochastic perturbations. All members show high values in the central Baltic Sea. The time series shown in Figure 6 represents the highest significant wave height on 22 February 2002 at 09 UTC. There is also a strong variability between the different ensemble members in this region. Wave heights in member 8 are especially higher than in the other members in the Gulf of Bothnia, but this member shows also much higher wave heights than the other members in the central Baltic Sea. In the western Baltic Sea the differences between the members are not that strong. The overall spatial pattern of the significant wave height looks similar between ERA5 and WRF ensemble members. The wave models (Fig. 6) driven by the WRF ensemble hindcast started from the ERA5 EDA initial conditions show a very small spread. A difference can be especially seen at the second peak. Much stronger differences are provoked by the WRF ensemble based on stochastic perturbations. Combining both ERA5 EDA fields as initial conditions and stochastical perturbations produces a spread of a similar size. The simulated significant wave heights of the most extreme members with about 11.2 m are clearly above the highest observations with about 8.2 m (Björkqvist

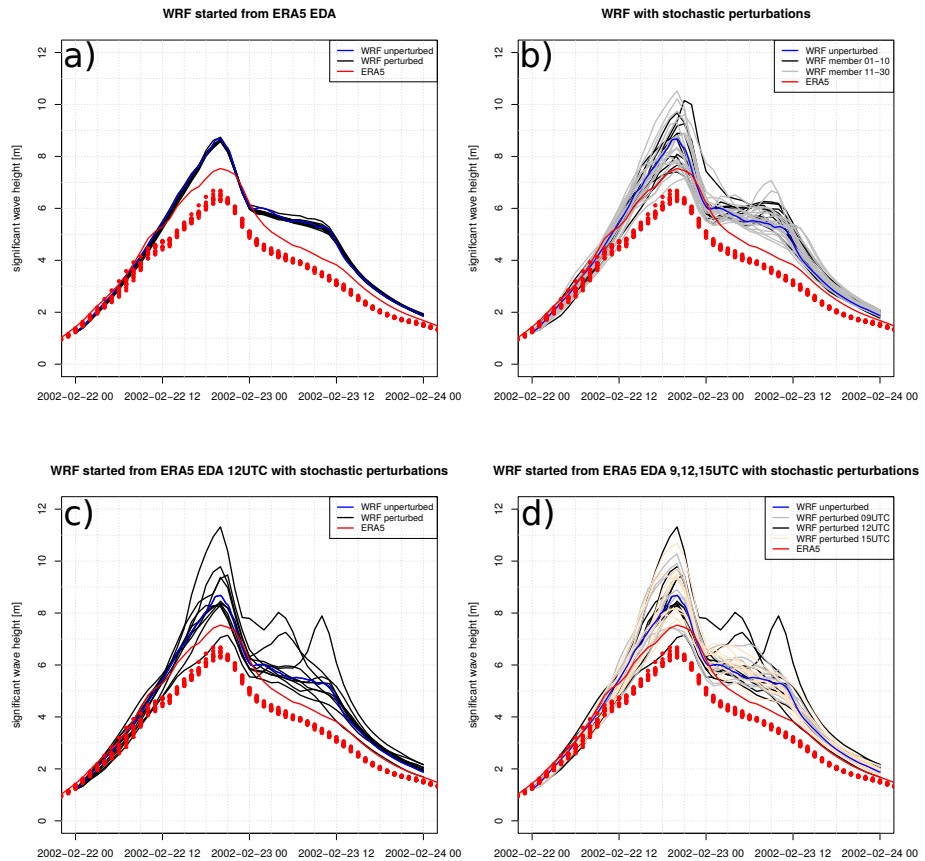

**Figure 6.** Significant wave height [m] at station Northern Baltic (21.00°E, 59.25°N, see Figure 1); driven by WRF ensemble based on a) initializations at 21 February 2002 12 UTC from ERA5 EDA, b) ERA5 reanalysis and stochastic perturbations, c) ERA5 EDA with SKEB and SPPT, and d) ERA5 EDA with SKEB and SPPT initialed additionally on same day at 09 and 15 UTC; ERA5 significant wave heigths from the ECWAM model in 0.36° resolution; Shown 22 February 2002 21 UTC.

et al., 2017). One reason could be an overdispersion of the wind fields of the WRF ensemble. The stochastic perturbations were not calibrated, as a larger number of hindcasted events are necessary to be able to optimize the perturbations. Another issue is the roughness length of the sea surface, which is defined as a constant value in the applied WRF setup. Under severe storm conditions, the roughness of the sea surface should increase, resulting in a reduction of atmospheric kinetic energy and

5   a corresponding limitation of wave growth. An investigation of the impact of the constant roughness of the sea surface on the wave height was out of the scope of this study. This effect could lead to a systematic overestimation of the wave heights in some storms. The storm events, Toini and Rafael, with the highest observed significant wave heights discussed by Björkqvist et al. (2017) were additionally hindcasted and are presented in the supplemental material. They seem to be less sensitive on the perturbations. Based on the short timeseries of observations, it is difficult to judge which significant wave height is still

10   realistic.

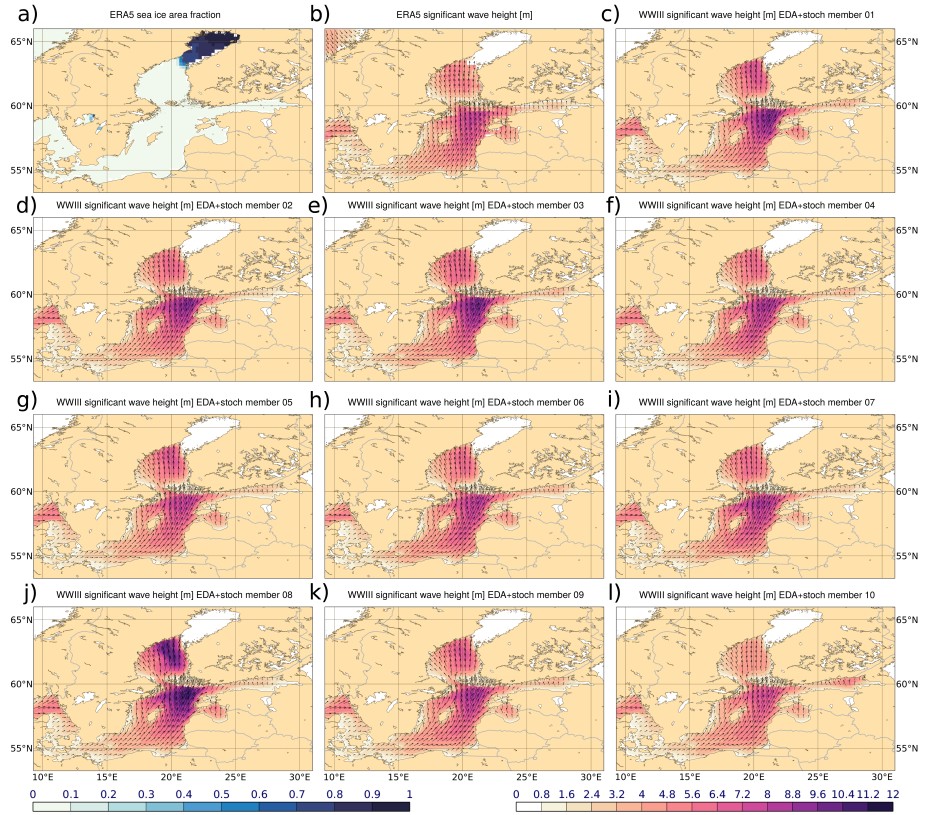

**Figure 7.** a) ERA5 sea ice area fraction [0-1]; b) ERA5 ECWAM significant wave height [m] with direction in meteorological convention and c-l) ten WAVEWATCH III® members driven by WRF ensemble initialised from ten ERA5 EDA members at 21 February 2002 12 UTC with SPPT and SKEB. Shown 22 February 2002 21 UTC.

A shortcoming of the presented procedure can be that the wave model runs were all started from the same initial state. This means that a certain time is needed until the different members diverge, especially as the total wave height is a combination of wind sea and swell. The later needs some time to travel, so that regions which are predominated by swell can be assumed to need a longer period to produce a reasonable spread with this setting. For the Baltic Sea, events with a strong influence of swell are infrequent (e.g. Broman et al., 2006; Soomere et al., 2012). French Guiana, for example, is one region which is swell dominated. Osinski et al. (2018) estimated the hundred-year return level of the significant wave height of northerly swell events at the French Guiana coast. Such events are generated in the Northern Atlantic and travel until the north-eastern coast of South America. For hindcasting such events with the demonstrated procedure, a large domain would be necessary and long lasting forecast horizons, so that the waves are already perturbed where generated and over their lifetime as well. This can lead to stronger deviations from the real past state.

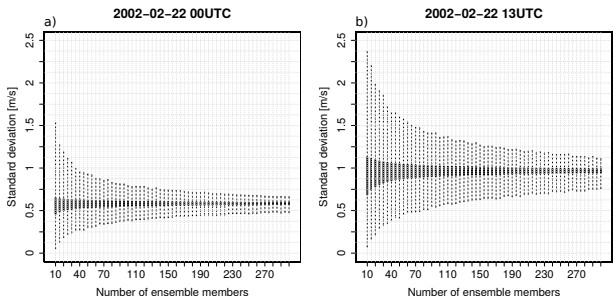

**Figure 8.** Box-whisker plots of the standard deviation of the 10 m wind speed at 19.39°E, 56.17°N of ten million samples of ensembles of size 10 to 300 randomly sampled from an ensemble with 500 members generated with WRF by applying SKEB and SPPT; shown a) 22 February 2002 00 and b) 13 UTC. Compare Fig. 2.

## 3.3 Robustness of the ensemble spread depending on the ensemble size

Each ensemble member is a random draw of the PDF of the forecast / hindcast uncertainty. In the extreme case of having only two members, it is very unlikely that the most extreme cases are represented. By increasing the ensemble size, the probability is getting higher that the full range of uncertainty is sampled. For local area models, operational weather forecast centres

produce ensembles with around ten to twenty members. At first sight, this number seems to be comparable to the presented study. If driving the regional model from a global ensemble with about 30 to 50 ensemble members, one can use a clustering technique to identify the most representative members instead of randomly selecting a small subsample, what improves the ensemble performance (e.g. Nuissier et al., 2012). If the ensemble is initialised several times per day, the different runs can be combined using the LAF approach (e.g. Raynaud and Bouttier, 2017). To predict the probability of the exceedance of a certain

threshold, one can apply also neighbourhood techniques (e.g. Theis et al., 2005) or post-processing techniques like Bayesian Model Averaging (Raftery et al., 2005). Neither the initial and lateral boundary conditions come from a large ensemble in this study nor the application of neighbourhood or other post-processing techniques helps, because the ensemble members are used to drive a wave model. To get an idea how many ensemble members are reasonable in this case, 500 members have been generated with stochastic perturbations. From these 500 members, an ensemble with N members is generated, with N starting

at 10 going until 300. Ten million samples of each ensemble of size N are selected by randomly choosing N out of the 500 members. The standard deviation is used as a measure for the ensemble spread and is calculated for each of the ten million samples of the ensemble of size N. The number of possible combinations of selecting N out of 500 members can be determined by using the binomial coefficient $\binom{500}{N}$. This number exceeds ten millions for all tested ensemble sizes between 10 and 300. If the ensemble size is reasonable to get a robust estimate of the uncertainty, the spread should be relatively similar between each

of the samples.

Figure 8 shows box-whisker plots for the ten million samples for ensemble sizes between 10 and 300 members. The variation of the spread in the ten million samples twelve hours after initialization is demonstrated in the left panel a). As it needs some time that the stochastic perturbations provoke spread between the ensemble members, there is a lead time dependence in the

spread. The right panel b) presents a situation 25 hours after initialization. At this time, the wind speed is very high, see Figure 2. In extreme situations, in which we are especially interested, we expect a higher uncertainty. This higher uncertainty is represented by a larger spread. All the ensembles with sizes between fifteen and hundred members show a median of the spread around one, at 22 February 2019 13 UTC. The ten member ensemble has a slightly lower median. With a higher uncertainty, a larger number of ensemble members is necessary to sample the entire uncertainty range. With increasing ensemble size, it is getting more probable that the entire uncertainty range is sampled. This is why the range of the box-whisker plots is decreasing with increasing ensemble size. At 22 February 2019 00 UTC, the uncertainty is lower and / or the spread as a measure of uncertainty is not yet fully developed after twelve hours. As the robustness of the ensemble spread seems to be dependent on the uncertainty, the range of the box-whisker plots is much inferior at 22 February 2019 00 UTC than thirteen hours later. To achieve a general statement about the ensemble size / spread relation, a much larger sample over a longer period must be investigated, but it can already be concluded that an ensemble size of only ten randomly generated members, as demonstrated in this application, can lead to a significant over- or underestimation of the uncertainty. Depending on the application, the ensemble size needs to be selected by a compromise between the robustness of the uncertainty estimate and the computational cost.

## 3.4 Impact of the spatiotemporal resolution of the atmospheric forcing on the significant wave height

The numerical time step of a wave model is less than a minute (typical for explicit numerical schemes) or few minutes (typical for implicit numerical schemes). The wave model therefore needs updated wind information e.g. every 30 seconds. This is done by interpolation from the wind forcing that is provided e.g. every hour or every third hour. A higher temporal resolution of atmospheric forcing data than one hour is normally not available. If a variable in the ocean-/wave model to be driven has a short response time (e.g. surface current generated by wind compared to sea surface temperature (SST) whose response is slower), and the variability of the atmospheric forcing in between the temporal resolution of the forcing fields is high, the result can be an under- or overestimation and an erroneous time evolution. One imaginable solution is to use maximum values during the output time interval of the atmospheric model, but this can lead to spatially inconsistent fields, especially if the time interval is very long. To test the impact of different temporal resolutions on the significant wave height, wind fields in 5 minutes resolution were produced with the 0.063° setup. Figure 9 shows the wind field in 5, 15, 30 and 60 minutes resolution at one specific grid cell and the resulting significant wave height at the same location and time. It can be seen that the wind speed maximum in the 60 minutes resolution is about 0.25 m/s below the maxima of the higher temporal resolutions. Between the higher temporal resolutions of the wind data, the wind speed maxima are very close. The effect between the 60 minutes temporal resolution and a forcing in higher temporal resolution on the significant wave height is relatively low with about 2 cm. Systematic differences cannot be found based on the small sample, but this sensitivity test indicates that the choice of the 15 minutes resolution is a reasonable compromise between a good representation of the extreme values and file size.

A stronger impact can be expected from the spatial resolution of the driving wind fields, because a coarser resolution of the atmospheric model can be assumed to produce lower extreme wind speeds as a grid cell represents the average value over the area it covers. By adapting, for example, the parameter betamax which describes the maximum value of wind-wave coupling,

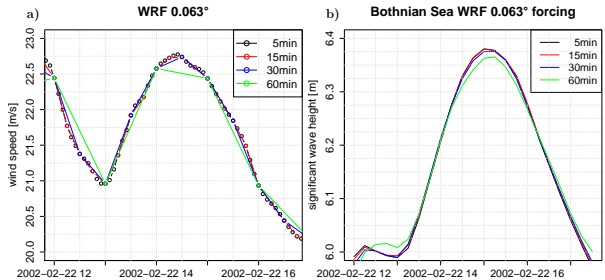

**Figure 9.** a) Wind speed [m/s] in 0.063deg setup and b) significant wave height [m] at 20.23°E and 61.8°N ; Testcase without sea ice

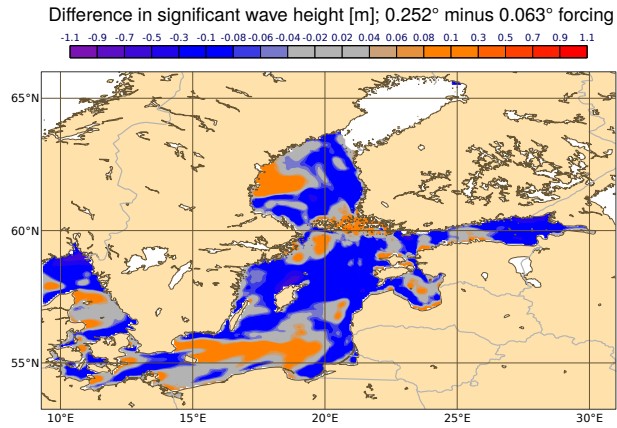

**Figure 10.** Difference in the significant wave height [m] in WAVEWATCH III® between simulation driven by WRF with 0.252° and by 0.063° in 15 minutes temporal resolution at 22 February 2002 21 UTC. The northern part of the Gulf of Bothnia is covered by sea ice.

this difference can be compensated for. A better representation of the complex coastline of the Baltic sea as well as of the various islands is given by the higher resolved WRF model. For this reason, a difference in the spatial pattern of the significant wave height can be expected. A test with the coarsest (0.252°) and the highest resolution (0.063°) produced in this study has been conducted. The same parameter sets were used, as a calibration is not possible based on the short period hindcasted with

5    WRF. Figure 10 shows the difference between these two forcings on one timestep in the significant wave height. One grid cell of the coarser WRF setup contains 16 grid cells of the high resolution setup. Wave height maxima as well as minima were found more extreme in the higher resolution with a higher spatial variability. It would be interesting to determine the remaining difference in the wave parameters provoked by the atmospheric forcings with different resolutions after a calibration of the wave model, done by applying an automatic and objective calibration procedure like for example the one proposed by Gorman

10    and Oliver (2018). Tuomi et al. (2014) studied the effect of different spatiotemporal resolutions of the wind forcing on a wave model with a higher spatial resolution than applied here. A wave model with a higher resolution might benefit more from a higher resolution of the wind forcing.

## 4    Conclusions

Different approaches for hindcasting a single relatively strong storm event in the Baltic Sea were tested in this study to create an ensemble hindcast of atmospheric (wind) and wave conditions based on a state-of-the-art atmospheric mesoscale model and a third generation wave model. The objective of the ensemble approach is a quantification of the uncertainty of the hindcast. The wave model was calibrated based on a publicly available regional reanalysis, and than validated with this dataset and also with forcing data produced with the atmospheric setup used in this study, as demonstrated in the supplemental material. A lagged-average WRF forecast ensemble showed only little spread with initial and lateral boundary conditions based entirely on the high resolution ERA5 reanalysis fields. The spread of the LAF ensemble can not be easily adapted.[15]

A domain shifting approach with ERA5, in which the input fields are shifted into all directions by one grid cell, shows a more or less similar spread to the LAF ensemble, with the same advantage of using the high resolution reanalysis data only. The number of ensemble members and the spread of the ensemble is limited to the number of reasonable shifts. Too large shifts can be expected to degrade the hindcast.[16]

Starting WRF from the ERA5 EDA members show also a spread of similar size as with the two other approaches. The disadvantage is the coarser resolution of the initial fields. Fine scale structures are not present in these fields so that the ensemble spread is limited. Stochastic perturbations produce a much larger spread, but need some time to develop. The first twelve hours are not used in this study, because they are assumed to be affected by a model spin-up. This is not a shortcoming for a hindcast procedure.

A combination of stochastic perturbations and an initialization from the ERA5 EDA fields produces also deviations from the unperturbed runs which are not present by only using the stochastic perturbations. This approach is especially interesting and is close to what is used in meterological weather forecast centres for the operational forecasts. The wind fields from this ensemble hindcast produce also a large spread in the wave model. A visual comparison with the ERA5 wave model ensemble of data assimilations indicates that this spread is more reasonable than the one obtained using the first three discussed ensemble generation approaches. The peak of the significant wave height in the Baltic proper of the most extreme members is, however, with about 11 m strongly exceeding existing observations in this region. One possible reason could be an overdispersion of the ensemble system. Another important factor is the roughness of the sea surface and its impact on the dynamics of the storm. In the presented setup, the roughness length of the sea surface is defined as a constant value. The constant sea surface roughness could lead to systematic overestimations of the wind speed resulting in too high wave heights. A coupling of the atmospheric

---

[15]A weighting of the different realizations (in this case of the wave model) by giving the runs more weight which are expected to have a lower error is possible. In this case more weight to the runs should be given which have smaller errors in the verification of a large sample of hindcasts (e.g. more weight to runs with lower lead time to the desired event), but this can be expected to not strongly enlarge the ensemble spread.

[16]As the WRF model has a finer resolution, shifts different than multiples of one grid cell by adding or substracting an offset onto the coordinates of the ERA5 grid would change the interpolation for the WRF initial and lateral boundary conditions. This was not tested, and it was also not investigated systematically if members generated from smaller shifts are closer to the unpertubed run or if shifts into a certain direction (e.g. into flow direction or perpendicular to it) lead to different spreads than shifts into other directions, which would mean that there are systematic differences between the members to be taken into account by the interpretation of the ensemble data. A test with shifts of two and three grid cells into north, west, south and east direction were tested and indicate that there are systematic differences.

with the wave model would allow to adapt the roughness length depending on the sea surfcace conditions and can lead to a limitation of the wave growth.

The robustness of the spread depending on the ensemble size was tested by randomly generating ensembles with different sizes (10 to 300 members) from an ensemble hindcast with 500 members. For small ensembles, the range of the ensemble spread can differ largely depending which members were randomly selected. In operational services, this problem is tackled by selecting for example already representative members from a larger global ensemble. To achieve a comparable robust estimate of the uncertainty, the ensemble size for the here presented approach without pre-selection of ensemble members must be larger than the one of operational local area model ensembles.

Another source of uncertainty arises from the spatiotemporal discretization of the atmospheric model and the resulting forcing fields for the wave model. Errors introduced by a coarse temporal resolution of the driving wind fields in the significant wave height are relatively small in this event testcase. For a strong event with a significant wave height of about 6.3 m, the difference in wave heights between simulations using 5 and 60 minutes temporally resolved wind forcing is only on the order of 2 cm. Between 15 minutes and 5 minutes temporal resolution, the impact on the wave height is negligible for the demonstrated case. The horizontal resolution has a much stronger impact. This can be potentially be corrected by calibrating the model to the different wind forcings. It would be interesting to estimate the remaining difference, but this was not possible in the framework of this study as a calibration of the models is not feasible based on a hindcast of only a single event.

A combination of ERA5 EDA fields as initial conditions and the stochastic perturbations showed the ability to produce a larger spread than with the other demonstrated approaches. Stochastic perturbations haven't been tuned in this study. Producing longer timeseries, for tuning and validating the model could lead to a reasonable measure of the hindcast uncertainty on the regional scale. Operational atmospheric and wave products exist with a comparable or even higher resolution than applied here, whose quality is superior to what can be reached with the demonstrated procedure as they include state-of-the-art data assimilation techniques. The application of such operational products is however limited by the available periods and also by the inhomogeneity of the datasets. The demonstrated approach allows to adapt the spatiotemporal resolution and the ensemble size to specific research questions for event based hindcasts in a homogeneous manner over the entire available ERA5 period.

*Code availability.* The WRF source code is available from https://github.com/wrf-model/WRF/releases and the WAVEWATCH III[®] from https://github.com/NOAA-EMC/WW3

*Data availability.* ERA5 and the UERRA/Harmonie-v1 reanalysis can be retrieved from the Climate data store at https://cds.climate.copernicus.eu.

*Sample availability.* Ensemble hindcasts of wind and wave fields presented in this study can be requeested by contacting the corresponding author.

*Author contributions.* RDO is responsible for the concept of this study, prepared the configurations of WRF and WAVEWATCH III®, conducted the simulations and prepared the presentation of the results. HR was involved in the discussion of the results and in the preparation of the manuscript.

*Competing interests.* The authors declare that there is no conflict of interest.

*Acknowledgements.* This study was financed by the Bonus Micropoll project, which has received funding from BONUS (Art 185), funded jointly by the EU and Baltic Sea national funding institutions. For the calibration and validation of the Baltic Sea WAVEWATCH III® setup, computing resources at the HLRN were consumed and E.U. Copernicus Marine Service Information were used. The simulations in this study were generated using Copernicus Climate Change Service Information (2018/2019). The research and work leading to the UERRA data set used in this study has received funding from the European Union Seventh Framework Programme (FP7/2007-2013) under grant agreement № 607193. We would like to thank the WRF and WAVEWATCH III® developers for providing their models over Github. We also would like to thank the two anonymous reviewers and the editor Judith Wolf for their comments which helped to improve the article.

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
