# Peer review of "Ensemble hindcasting of wind and wave conditions with WRF and WAVEWATCH III® driven by ERA5"

_Ocean Science, 2019_

## Referee Comment (RC1) · Anonymous Referee #1 · 23 Aug 2019

Review comments for "Ensemble hindcasting of wind and wave conditions with WRF and Wavewatch III ® driven by ERA5" (os-2019-76) by Robert Daniel Osinski and Hagen Radtke

The authors present a results where different methods to produce ensemble wind hindcasts (to be used for wave hindcasts) are compared. They conclude that varying the starting time of the hindcast/forecast, or a single grid cell shifting, leads to quite a small spread. The stochastic perturbation of the ERA5 EDA field lead to the largest spread in the ensemble.

General comments: I like this paper, and I can recommend it for publication after some

minor alterations. It is within the scope of Ocean Science, the topic is of interest for the scientific community, it is timely, and well executed. It is obvious that the authors have a solid expertise in atmospheric modelling and ensemble forecast compilation. It does, however, seem that they might not bee completely familiar with wave modelling efforts that has already been done in the Baltic Sea. This doesn't diminish the relevance of this paper, but I will provide a few references at the end. Not all of these need to be included in the paper, nor is it meant to be exhaustive. They are more intended as a friendly starting point to aid in finding some relevant research to put this study into better context.

Major comments:

Major comment #1) The introduction is well written from the point of view of ensemble modelling, but it totally lacks material on Baltic Sea waves and the relevant research. Please see my list of references in the end as a starting point. Also the discussion of the results needs to be tied better to what we as a community know about Baltic Sea wave conditions.

Major comment #2) While I can get on board with using only one storm in this study, I think it is very unfortunate that the authors have chosen the 2002 storm when no data from the NBP wave buoy is available. For example the 2004 Rafael storm would have wave buoy data available for validation. It might be unreasonable to redo the model runs (I will leave that to the authors), but at least the authors should discuss how realistic the highest values (Hs>11 m) are by comparing to what we know about the Baltic Sea wave climate (again, see the list of references at the end).

Specific comments:

**1) The wave model is "WAVEWATCH III", not "Wavewatch III".**

**2) page 1 line 25:**

Perhaps have a paragraph break at "In principle"?

**3) page 3 line 23 "The ERA5 dataset was used in this study to drive the atmospheric model WRF, a coarse Wavewatch III ® wave model to provide lateral boundary conditions for a Wavewatch III ® wave model with higher resolution and for comparison with the model results."**

This is a bit unclear and should perhaps be rewritten.

**4) page 5 line 3 "UERRA/Harmonie-v1 was used for calibration and validation of the setup against one month of data from buoys available from the Copernicus Marine environment monitoring service 12 (CMEMS) with the previous Wavewatch III v5.16 version."**

This is unclear. What buoy data exactly was taken from the CMEMS database? Was the validation done with a different version of WAVEWATCH III than the actual results? What does "calibration" mean? Was some of the source terms calibrated specifically for the Baltic Sea? Was the validation metrics similar to those of previous modelling efforts in the Baltic Sea?

**5) Please add some kind of Table of the different type of ensembles. As written, it is a bit hard to follow.**

**6) Fig 2: "results shown at 19.39°E, 56.17°N"**

Show this point in Fig. 1.

**7) Fig 3.**

There are a lot of subplot. Would it be sufficient to just use max difference to the mean, or to reduce the number of panels in some other way?

**8) page 10 line 24: A shortcoming of this procedure**

A bit unclear what is meant by "this"

**9) page 11 lines 1-2:**

Baltic Sea not really swell dominated, so this shouldn't be an issue in your results, and the discussion seems a bit off key, especially in the middle of the paper concentrating on the Baltic Sea. It is up to the authors if they want to keep it. Just thought I would point out how it looks from a Baltic Sea perspective.

**10) page 13 line 2.**

Perhaps start a new paragraph with "Figure 8 shows..."?

**11) page 13 line 19 "The time step of a high resolution ocean or wave model is normally below one hour."**

This is slightly misleading, since one hour is a typical time resolution for the output of a wave model. The time step of a wave model can be counted in seconds (typical for explicit numerical schemes) or minutes (typical for implicit numerical schemes). The wave model therefore need updated wind information e.g. every 30 seconds. This is done by interpolation from the wind forcing that is provided e.g. every hour or every third hour.

**12) page 14 line 9-10: "Systematic differences cannot be found based on the small sample, but it indicates that the choice of the 15 minutes resolution is a reasonable compromise between a good representation of the extreme values and file size."**

I think one could argue that a 60 minute resolution is reasonable, since a difference of 2 cm is under 1%. This is small compared to the sampling variability (roughly 5-10%) that is present in measured significant wave height data that we routinely use to validate the models. Still, 15 minutes is clearly also a reasonable choice, so I'm not arguing with that part of your conclusion.

**13) page 14 line 15-16: "For this reason, a difference in the spatial pattern can be assumed."**

Do you mean that a difference can be expected?

**14) last paragraph on page 14:**

It think it is worth noting that the operational products typically used to force Baltic Sea wave models are already close to the higher resolution (0.063 deg). While this sensitivity test is very welcome, it could easily be read as if the wave modelling community is currently using insufficient wind forcings is no context is provided. It might also be worth noting, that separate high-resolution wave model implementations might benefit more from higher resolutions in the wind forcing than what is seen in a 1 nmi Baltic Sea wide wave model. This kind of sensitivity tests for coastal wave models have been done in the Baltic Sea (see e.g. Tuomi et al., 2014).

**15) page 16 line 1-2: "As the first twelve hours are not used, because of the model spin-up, this is not really a shortcoming."**

This will not be true for operational wave forecasts that get their starting conditions from the previous run. Will it be a shortcoming then?

**16) page 16 lines 11-13 "To achieve a comparable robust estimate of the uncertainty, the ensemble size for the here presented approach must be larger than the one of operational local area model ensembles."**

Just to be clear, is the "here presented approach" choosing the members at random? In other words, is your conclusion that choosing random members requires more members in the ensemble than if they are "screened" in advance using a coarse model, or are you trying to make some additional point?

**17) page 16 line 16-17: "For a strong event, the difference between a 5 and 60 minutes temporally resolved wind forcing is only on the order of 2 cm."**

I think it is a bit questionable to give an absolute difference without knowing the significant wave height. This doesn't really provide that much useful information.

**18) In e.g. Figure 2: are you using the wave product of ERA5, or are you using WAVEWATCH III forced with ERA5 winds?**

**19) If you are only simulating the wave field in the Baltic Sea, then there is not really a need to nest it outside of the Danish straits, since no significant amount of wave energy will penetrate. It's not wrong, just pointing out that it is not really necessary.**

**20) The figures are sometimes very hard to read. Please prepare them according to the guidelines of the journal (fonts sizes, labeling of subpanels etc.)**

List of (potential) references:

Tuomi, L., Pettersson, H., Fortelius, C., Tikka, K., Björkqvist, J.-V., and Kahma, K. K., 2014: Wave modelling in archipelagos. Coastal Engineering 83, pp. 205-220. https://doi.org/10.1016/j.coastaleng.2013.10.011

Coastal wave modelling in the Baltic Sea with different atmospheric models using different resolutions and time steps.

Björkqvist, J.-V., Tuomi, L., Tollman, N., Kangas, A., Pettersson, H., Marjamaa, R., Jokinen, H., and Fortelius, C., 2017: Brief communication: Characteristic properties of extreme wave events observed in the northern Baltic Proper, Baltic Sea, Nat. Hazards Earth Syst. Sci., 17, pp. 1653-1658, https://doi.org/10.5194/nhess-17-1653-2017

Comparing wave forecasts with different lead time against wave buoy measurements from a storm.

Soomere, T., Behrens, A., Tuomi, L., and Nielsen, J. W.: Wave conditions in the Baltic Proper and in the Gulf of Finland during windstorm Gudrun, Nat. Hazards Earth Syst. Sci., 8, 37-46, https://doi.org/10.5194/nhess-8-37-2008, 2008.

Comparing different wave forecasts against measurements from a storm.

Tuomi, L., Kahma, K. K., and Pettersson, H.: Wave hindcast statistics in the seasonally ice-covered Baltic Sea, Boreal Environ. Res., 16, 451–472

Basic wave statistics for the Baltic Sea.

Olga Vähä-Piikkiö, Laura Tuomi, Vibeke Huess, 2019, Baltic Sea Wave Analysis and Forecasting Product BALTICSEA_ANALYSIS_FORECAST_WAV_003_010,http://cmems-resources.cls.fr/documents/QUID/CMEMS-BAL-QUID-003-010.pdf

Information about a currently running wave forecast

Nikolkina, I., Soomere, T., and Räämet, A.: Multidecadal ensemble hindcast of wave fields in the Baltic Sea. In: The 6th IEEE/OES Baltic Symposium Measuring and Modeling of Multi-Scale Interactions in the Marine Environment, May 26–29, Tallinn Estonia. IEEE Conference Publications, 9 pp., https://doi.org/10.1109/BALTIC.2014.6887854, 2014.

WAM simulations in the Baltic Sea with different wind products

―――――――――――――――――――

---

## Referee Comment (RC2) · Anonymous Referee #2 · 30 Aug 2019

This manuscript provides an interesting insight into possibilities of the construction of a large ensemble of hindcasts of wave properties in the Baltic Sea region. On the one hand, this approach is thought-provoking in itself as the pool of similar studies is very limited in this area. On the other hand, it is not clear beforehand how large is the potential of this approach to improve the hindcast as most of the discrepancies of the wave field reconstructions seem to stem from uncertainties of the driving wind fields. In particular, even small variations in the trajectories of low pressure systems may lead to large changes in the wave properties in the study area. It is thus important to understand how the possible uncertainties in wave reconstruction can be "distributed" between the variations in the driving fields and the specific ways of the description of

wave physics. The topic thus clearly fits the scope of Ocean Science.

It is a pity that the approach is applied to an event in February 2002 for which essentially no ground truth about wave properties is available in the area of high waves. While the wave buoy of the Finnish Meteorological Institute was removed because of possible ice impact, the bottom-placed device at Almagrundet (Broman et al., 2006) did not provide any data in February 2002. However, as it is said both in Abstract and Conclusions that the event "provoked a severe storm surge in February 2002" it is necessary include at least some numbers and locations to substantiate this information. For example, nothing specific happened in Latvian waters.

The method for the construction of the ensemble is rational and interesting. It is reasonable from the viewpoint of wind fields but seems to run into problems in terms of wave properties. It is of course worth of trying to construct as large ensemble as possible in order to examine the spread. However, it is not a good sign that some members of the ensemble lead to unrealistic wave heights. Both Fig. 2 and Fig. 3 indicate that maximum wind speeds in the northern Baltic proper are mostly in the range of 20–22 m/s and only for a few members reach the level of 25 m/s. Such winds speeds only cover a small part of the northern Baltic Proper. Even though the wind direction was favorable for the generation of high waves in this area, it is unlikely that significant wave heights substantially exceeded 7 m in this storm. Wave heights exceeding 8 m are very infrequent in this region. Even in the extreme storm Gudrun/Erwin (January 2005, 10-min wind speed >28 m/s in large sea areas) wave heights most likely did not exceed 10 m anywhere in the Baltic Sea (Soomere et al., 2008).

Therefore, I guess that wave heights between 11 and 12 m in Fig. 6 are completely unrealistic for the February 2002 storm. It seems that the entire ensemble severely (by almost 2 m on average) overestimates wave heights in the northern Baltic proper. Thus, I recommend to extensively comment this feature and to include a short insight into measured or modelled wave heights in this area for storms of comparable properties. Ideally, I would recommend to include a paragraph or two about extreme wave

properties in the study area, following either (Tuomi et al., 2011) or (Björkqvist et al., 2018).

In particular, I recommend extending the message on page 5, line 5–6 towards a sound explanation that the model is essentially uncalibrated for the Baltic Sea conditions. This is mentioned in the last sentence before conclusions on page 14. The point of this sentence should be made very clear from Abstract to Conclusions. I stress that such a bias in the evaluated wave heights does not undermine the validity of most of the results but it should be made clear to the reader that single values of wave height (and even the ensemble average) do not necessarily match the wave properties in this storm.

For the listed reasons I recommend moderate to major modifications to the manuscript. It is essential that the reader is informed (i) about some basic features of wave climate and extreme waves in the Baltic Sea and also (ii) that the simulations probably strongly overestimate wave heights and (iii) are performed specifically to study the spreading properties of ensembles, with no exact relevance to the actual wave heights during the simulation interval. An absolute must is to inquire the modelled data from a properly calibrated run (e.g., from the authors of Björkqvist et al., 2018) for the underlying location of Fig. 6 to give a minimum flavor of the possible bias.

The text is written in fairly good English but reveals slight German accent in the form of very long sentences at places and missing of some articles in the text. It is mostly clear but still needs extensive polishing, especially closer to the end of the manuscript. As I am not native speaker, I only include a list of clear typos below.

Minor comments

The paragraphs are at places very long. For example, the first paragraph of Introduction extends over 28 lines. It is recommended to split long paragraphs into shorter ones.

The style of calendar days ("21. February 2002" on page 6, line 4 and "22nd to 24th of

February" on the next line) should be unified.

The first two sentences of Abstract seem unnecessary.

Page 1, line 17: probably should be "and is described".

Line 23 and some other locations: some journals require comma after "e.g."

Page 2, lines 32–34: the sentence does not make sense; possibly because of too strong German accent.

Page 3, line 13: C3S has already been explained on page 2, line 22.

Line 20: probably full stops are not necessary in "21. February 2002" and similar expressions.

Line 23 it is better to say that 0.36deg and 1deg denote the resolution of the relevant grid. Please do so also in several locations below where the size in degrees is given without any explanation.

Page 4, line 10: please specify the meaning of "writing 15 minutes output".

Line 12: please explain what is meant under "the temporal impact" (probably the dependence of the solution on the time step).

Line 17: please specify the meaning of "Eta layers".

Line 18–19: consider replacing the jargon-like expression "until fine scales develop" by a more explanative one. Please do so also in several occasions below to avoid clash in the meaning of, e.g., "finer scales are not represented" on page 6, line 9.

Page 5, line 1: to avoid misinterpretation, I suggest to mention that nesting of the wave model to the Baltic Sea is not really necessary for the hindcast of wave properties in the central and northern regions of this water body because very little wave energy penetrates through the Danish straits.

The reasoning on lines 2–6 is only partially relevant for the conditions of the Baltic Sea.

Line 7: while most of the model setup is obviously fine for the Baltic Sea, please comment on the adequacy of the use of the chosen frequency range for this water body. Wave modellers usually substantially extend the frequency space here. The team of the Finnish Meteorological Institute normally uses 35 frequencies (Laura Tuomi et al., many papers) and some research in subbasins of the Baltic Sea even 42 frequencies (0.0418–2.08 Hz, Soomere, 2005). It is probably not necessary to cover such an extended range. However, insufficient coverage of short waves may lead to too slow wave growth under rapidly increasing wind conditions.

Lines 14–16: the message of the entire sentence is technically clear but seems misplaced or even irrelevant.

Page 7, line 7: "these".

Page 10, lines 19–20, the sentence "Compared to ERA5, the overall spatial pattern is comparable" does not make sense to me.

Page 11, lines 2–5: the reasoning is almost irrelevant for the Baltic Sea conditions and should be left out. Instead, it should be emphasized that strong swells are infrequent in the Baltic Sea (see, e.g., Broman et al., 2006; Soomere et al., 2012) and thus deviations in the hindcast or forecast driven by the accuracy of the representation of swells are usually not very large in this water body.

Page 12, line 14: something is wrong with "500 choose N possibilities exist".

Page 13, line 10–12: the sentence is unclear.

Line 13: "developed"; also, the entire sentence remains partially unclear starting from "why".

Lines 16–17: the concluding sentence of the subsection should be made clearer.

Line 18: use "on" instead of "onto".

Page 14, line 7: please specify what is meant under "The higher temporal resolutions

do not differ so much." Also, the subsequent sentences contain too much jargon.

Line 14: "orography of the coastlines" sounds weird as the height of the coastline is just zero; also: use "Baltic Sea".

Line 15: spatial pattern of what?

Line 5 or another appropriate place: please stress that an uncalibrated (for the Baltic Sea conditions) wave model was used but still the results about the spread are valid.

Page 16, line 1: remove "by this fact".

Lines 1–2: the message of the sentence "As the first twelve hours are not used, because of the model spin-up, this is not really a shortcoming." remains unclear.

Line 14: correct "atmopsheric".

Line 20: correct "possbile".

References:

Björkqvist, J.V., Lukas, I., Alari, V., van Vledder, G.P., Hulst, S., Pettersson, H., Behrens, A., Männik, A. 2018. Comparing a 41-year model hindcast with decades of wave measurements from the Baltic Sea. Ocean Engineering, 152, 57–71, doi: 10.1016/j.oceaneng.2018.01.048

Broman, B., Hammarklint, T., Rannat, K., Soomere, T., Valdmann, A. 2006. Trends and extremes of wave fields in the north–eastern part of the Baltic Proper. Oceanologia, 48 (S), 165–184.

Soomere, T. 2005. Wind wave statistics in Tallinn Bay. Boreal Environment Research, 10, 103–118.

Soomere, T., Behrens, A., Tuomi, L., Nielsen, J.W. 2008. Wave conditions in the Baltic Proper and in the Gulf of Finland during windstorm Gudrun. Natural Hazards and Earth System Sciences, 8, 37–46.

Soomere,T., Weisse, R., Behrens, A. 2012. Wave climate in the Arkona Basin, the Baltic Sea. Ocean Science, 8(2), 287–300, doi: 10.5194/os-8-287-2012.

Tuomi, L., Kahma, K.K., Pettersson, H. 2011. Wave hindcast statistics in the seasonally ice-covered Baltic Sea. Boreal Environmental Research, 16, 451–472.

---

## Author Comment (AC1) · 13 Sep 2019

Dear reviewer #1,

Thank you for your review and your comments. Additional supplemental material was prepared and uploaded regarding the calibration/validation procedure of the WWIII model and ensemble hindcasts of the storms Rafael and Toini. Please find in the following answers to your comments.

[Figure]

**1   Major comments:**

*#1) The introduction is well written from the point of view of ensemble modelling, but it totally lacks material on Baltic Sea waves and the relevant research. Please see my list of references in the end as a starting point. Also the discussion of the results needs to be tied better to what we as a community know about Baltic Sea wave conditions.*

Thank your for the list of publications concerning wave conditions in the Baltic Sea. We will include additional information and citations to existing studies in the introduction of the article. Following the article of Björkqvist et al. (2017) we added ensemble hindcasts of two additional storm events in the supplemental material and test another discretization of the energy spectrum as proposed by reviewer #2, following Soomere (2005). This will be discussed in the article. We also add a short paragraph on Baltic Sea wave climate to the introduction.

*#2) While I can get on board with using only one storm in this study, I think it is very unfortunate that the authors have chosen the 2002 storm when no data from the NBP wave buoy is available. For example the 2004 Rafael storm would have wave buoy data available for validation. It might be unreasonable to redo the model runs (I will leave that to the authors), but at least the authors should discuss how realistic the highest values (Hs>11 m) are by comparing to what we know about the Baltic Sea wave climate (again, see the list of references at the end).*

Storm Rafael and Toini were additionally hindcasted with the newest setup (please see the supplemental material). WWIII was calibrated on basis of the UERRA/Harmonie-v1 wind data and gives a satisfactory performance (please compare the supplemental material). It is also shown that the wave heights for the two additional storms (Rafael and Toini) with both WRF-ARW and UERRA/Harmonie-v1 show realistic wave heights.

For this reason, we assume that the significant wave height for the 2002 event with the unperturbed WRF-ARW wind forcing is also realistic. The perturbations of the WRF-ARW model physics were not tuned. To be able to do this, several extreme events would have to be hindcasted. One reason for the extreme wave heights in some ensemble members could therefore be an overdispersion of the wind fields from the WRF-ARW ensemble. In our WRF-ARW setup, the roughness length over the sea is assumed to be constant. Under severe storm conditions the sea surface roughness should increase with an effect on the wind field resulting in a limitation of the wave growth. A coupled WRF-WWIII setup would take this into account. By comparing the EPSgrams from the ECMWF (see for example ECMWF presentation [1] slide 20), one can see that the range of uncertainty can be very large. Based on a limited number of observations of extreme wave heights, it is therefore hard to judge which significant wave height is still realistic. We will discuss this in the article.

**2 Specific comments:**

*#1) The wave model is "WAVEWATCH III", not "Wavewatch III"*
This will be changed.

*#2) page 1 line 25: Perhaps have a paragraph break at "In principle"?*
A paragraph break is added there.

*#3) page 3 line 23 "The ERA5 dataset was used in this study to drive the atmospheric model WRF, a coarse Wavewatch III wave model to provide lateral boundary conditions for a Wavewatch III wave model with higher resolution and for comparison with the*
* * *
[1]https://confluence.ecmwf.int/download/attachments/55116817/OCEAN_WAVE_FORECASTING_AT_ECMWF_version_201602.pdf?api=v2

*model results." This is a bit unclear and should perhaps be rewritten.*
Changed to: "ERA5 is used for the initial and lateral boundary conditions for the atmospheric hindcasts with the WRF model. Lateral boundary conditions for the Baltic Sea WAVEWATCH III setup originate from a setup for the North Sea. This coarser model is driven by ERA5 winds. ERA5 reanalysis and EDA data are used for comparison of the hindcasts produced with WRF and WAVEWATCH III."

*#4) page 5 line 3 "UERRA/Harmonie-v1 was used for calibration and validation of the setup against one month of data from buoys available from the Copernicus Marine environment monitoring service 12 (CMEMS) with the previous Wavewatch III v5.16 version."*
Information to these questions is added in detail in form of supplemental material. At this part of the article, we will refer to this supplemental material. We see this article more as a demonstration of a principle idea for an ensemble hindcast procedure. For this reason, we think that it is sufficient if these details are presented as a supplement.

*#5) Please add some kind of Table of the different type of ensembles. As written, it is a bit hard to follow.*
Will be added.

*#6) Fig 2: "results shown at 19.39°E, 56.17°N". Show this point in Fig. 1.*
Will be added.

*#7) Fig 3.There are a lot of subplot. Would it be sufficient to just use max difference to the mean,or to reduce the number of panels in some other way?*
Figure 3 includes only the ensemble mean, minimum and maximum of the different ensemble generation approaches. Only the difference to the mean would neglect the

fact that the spread cannot be assumed to be symmetric around the mean. Figure 4 includes also a lot of subpanels. This presentation called postage stamps, is often used to present ensemble forecasts. For this reason, we prefer to keep in this way.

*#8) page 10 line 24: A shortcoming of this procedure ... .A bit unclear what is meant by "this"*
Changed to: "A shortcomig of the presented procedure for the wave hindcasts ..."

*#9) page 11 lines 1-2: Baltic Sea not really swell dominated, so this shouldn't be an issue in your results, and the discussion seems a bit off key, especially in the middle of the paper concentrating on the Baltic Sea. It is up to the authors if they want to keep it. Just thought I would point out how it looks from a Baltic Sea perspective.*
ERA5 is a global reanalysis. This is why the presented procedure for ensemble hindcasting can be applied for any region in the world. For this reason, we mentioned this point.

*#10) page 13 line 2.Perhaps start a new paragraph with "Figure 8 shows..."?*
We will start there a new paragraph as suggested.

*#11) page 13 line 19 "The time step of a high resolution ocean or wave model is normally below one hour."This is slightly misleading, since one hour is a typical time resolution for the output of a wave model. The time step of a wave model can be counted in seconds (typical for explicit numerical schemes) or minutes (typical for implicit numerical schemes). The wave model therefore need updated wind information e.g. every 30 seconds. This is done by interpolation from the wind forcing that is provided e.g. every hour or every third hour.*
We will adapt this part to: "The numerical time step of a wave model can be counted
in seconds (typical for explicit numerical schemes) or minutes (typical for implicit numerical schemes). The wave model therefore need updated wind information e.g. every 30 seconds. This is done by interpolation from the wind forcing that is provided e.g. every hour or every third hour."

*#12) page 14 line 9-10: "Systematic differences cannot be found based on the small sample, but it indicates that the choice of the 15 minutes resolution is a reasonable compromise between a good representation of the extreme values and file size."I think one could argue that a 60 minute resolution is reasonable, since a difference of 2 cm is under 1%. This is small compared to the sampling variability (roughly 5-10%) that is present in measured significant wave height data that we routinely use to validate the models. Still, 15 minutes is clearly also a reasonable choice, so I'm not arguing with that part of your conclusion.*
We agree that 60 minutes is reasonable. We only wanted to demonstrate that there might be an impact if using a higher temporal resolution. Of course, in the demonstrated case it is very small.

*#13) page 14 line 15-16: "For this reason, a difference in the spatial pattern can beassumed. "Do you mean that a difference can be expected?*
We change this to "expected".

*#14) last paragraph on page 14: It think it is worth noting that the operational products typically used to force Baltic Sea wave models are already close to the higher resolution (0.063 deg). While this sensitivity test is very welcome, it could easily be read as if the wave modelling communityis currently using insufficient wind forcings is no context is provided. It might also beworth noting, that separate high-resolution wave model implementations might benefit more from higher resolutions in the wind forcing than what is seen in a 1 nmi BalticSea wide wave model. This kind of sensitivity tests for*

*coastal wave models have been done in the Baltic Sea (see e.g. Tuomi et al., 2014).*
This study has been done from the perspective of a research institute rather than an operational forecast centre. We are aware and mentioned it also in the manuscript that an operational product should be of higher quality then what we are able to do with this setup. As a research institution, we often do not have access or cannot rely on operational datasets only, since we are interested in hindcasting events over a long period as determined by the research question. We would be limited by applying operational products regarding to the available periods, but also in terms of homogeneity of the dataset, which is required for investigations of long-term changes. With ERA5 as a global reanalysis and the atmospheric and wave models available from github, we demonstrate an approach, which everybody could repeat for any region in the world. When ECMWF extends ERA5 back to 1950, nearly 70 years of data are available for the production of event based hindcasts in a homogeneous way. One very relevant question is then which resolution is neccessary and how large should be the ensemble for the hindcasts. Should we produce more members or do we get more benefit from a higher resolution ? We tried to discuss these issues in the article. The ensemble runs were also done here in a coarser resolution than 0.063 deg, because it would have delayed the study because of computational limits. We will include this point about the impact of higher wind field resolution on higher resolved wave models and will make it clear that the point of a refined horizontal resolution applies to hindcasts rather than operational applications.

*#15) page 16 line 1-2: "As the first twelve hours are not used, because of the model spin-up, this is not really a shortcoming." This will not be true for operational wave forecasts that get their starting conditions fromthe previous run. Will it be a shortcoming then?*
We use a reanalysis from a different model and coarser resolution as the WRF model. In an operational setup, one would probably use data assimilation which combines the background from a previous model run based on the same model with the same

parametrisations with actual observations. This should reduce a spin-up significantly. There are other techniques to reduce the spin-up, also mentioned in the article, like Digital Filter Initialization for example. The spread develops also over the forecast horizon, why there might be a lack of spread during the first hours. This can be improved by applying an ensemble data assimilation technique.

*#16) page 16 lines 11-13 "To achieve a comparable robust estimate of the uncertainty, the ensemble size for the here presented approach must be larger than the one of operational local area model ensembles. "Just to be clear, is the "here presented approach" choosing the members at random? In other words, is your conclusion that choosing random members requires more members in the ensemble than if they are "screened" in advance using a coarse model, or are you trying to make some additional point?*
With the presented approach, the ensemble size must be larger than in case of pre-selecting already a representative subsample of ensemble members, because the ensemble members are generated in a random way in terms of the stochastic perturbations. We will be more specific: "The here presented approach without pre-selection of ensemble member ..."

*#17) page 16 line 16-17: "For a strong event, the difference between a 5 and 60 minutes temporally resolved wind forcing is only on the order of 2 cm. "I think it is a bit questionable to give an absolute difference without knowing the significant wave height. This doesn't really provide that much useful information.*
The significant wave height of about 6.3m will be mentioned here.

*#18) In e.g. Figure 2: are you using the wave product of ERA5, or are you using WAVEWATCH III forced with ERA5 winds?*
We tested also the ERA5 wind as forcing data and found a relatively good model

performance with an underestimation of the extreme wave heights in WWIII. For comparison, we showed the significant wave height from the ERA5 ECWAM with about 0.36° resolution (Fig. 6 and 7) and the ERA5 ECWAM uncertainty measure with about 1° resolution (Fig. 6). We will make this clearer.

*#19) If you are only simulating the wave field in the Baltic Sea, then there is not really aneed to nest it outside of the Danish straits, since no significant amount of wave energywill penetrate. It's not wrong, just pointing out that it is not really necessary.*
Our later application of the ensemble data are transport simulations with an ocean model for which we use the ensemble wave and atmospheric data as input fields. As we want to have also realistic wave parameters north of the Danish Straits, we used the presented nesting procedure.

*#20) The figures are sometimes very hard to read. Please prepare them according tothe guidelines of the journal (fonts sizes, labeling of subpanels etc.*
We will adapt the figures.

Please also note the supplement to this comment:
https://www.ocean-sci-discuss.net/os-2019-76/os-2019-76-AC1-supplement.pdf

---

## Author Comment (AC2) · 13 Sep 2019

Dear reviewer #2,

Thank you for your review and your comments. Additional supplemental material was prepared and uploaded regarding the calibration/validation procedure of the WWIII model and ensemble hindcasts of the storms Rafael and Toini. Please find in the following answers to your comments.

[Figure]

**1 Major comments:**

*This manuscript provides an interesting insight into possibilities of the construction of a large ensemble of hindcasts of wave properties in the Baltic Sea region. On the one hand, this approach is thought-provoking in itself as the pool of similar studies is very limited in this area. On the other hand, it is not clear beforehand how large is the potential of this approach to improve the hindcast as most of the discrepancies of the wave field reconstructions seem to stem from uncertainties of the driving wind fields. In particular, even small variations in the trajectories of low pressure systems may lead to large changes in the wave properties in the study area. It is thus important to understand how the possible uncertainties in wave reconstruction can be "distributed" between the variations in the driving fields and the specific ways of the description of wave physics. The topic thus clearly fits the scope of Ocean Science.*

*It is a pity that the approach is applied to an event in February 2002 for which essentially no ground truth about wave properties is available in the area of high waves. While thewave buoy of the Finnish Meteorological Institute was removed because of possible ice impact, the bottom-placed device at Almagrundet (Broman et al., 2006) did not provide any data in February 2002. However, as it is said both in Abstract and Conclusions that the event "provoked a severe storm surge in February 2002" it is necessary include at least some numbers and locations to substantiate this information. For example, nothing specific happened in Latvian waters.*

Concerning the first remark about the applicability of this approach, it has to be mentioned that there is especially an interest from the insurance sector to produce large samples of historical events to get a more robust estimate of, for example, the 200 year return level as defined by the Solvency II directive. Often statistical methods are applied to enlarge the samples producible from datasets like reanalysis. Osinski et al. (2016) used the archive of EPS forecasts from the ECMWF to produce an enlarged ensemble of historical events. The problem with operational forecasts is the inhomogeneity and limited period. With our approach, ensemble hindcasts back to

1979 (eventually 1950 if ECMWF extends ERA5) can be created in a homogenous way. Our later application is a simulation of particle transport with an ocean model and a study of the impact of the metocean uncertainty on the transport pattern and amount of material.

Regarding your second remark about the missing observations, two storm events (Rafael and Toini) were hindcasted additionally. Information about the calibration procedure, validation of the model and the presentation of the two storm events is added in form of supplemental material. The results of the 2002 storm event were compared in the article to ERA5 wind and wave data. Based on a single event, it is not possible to judge if the ensemble spread is reasonable. For this reason, we compared it with the uncertainty measure provided with the ERA5 reanalysis to get a rough idea about it.

*The method for the construction of the ensemble is rational and interesting. It is reasonable from the viewpoint of wind fields but seems to run into problems in terms ofwave properties. It is of course worth of trying to construct as large ensemble as possible in order to examine the spread. However, it is not a good sign that some membersof the ensemble lead to unrealistic wave heights. Both Fig. 2 and Fig. 3 indicate thatmaximum wind speeds in the northern Baltic proper are mostly in the range of 20–22m/s and only for a few members reach the level of 25 m/s. Such winds speeds onlycover a small part of the northern Baltic Proper. Even though the wind direction wasfavorable for the generation of high waves in this area, it is unlikely that significant waveheights substantially exceeded 7 m in this storm. Wave heights exceeding 8 m are veryinfrequent in this region. Even in the extreme storm Gudrun/Erwin (January 2005, 10-min wind speed >28 m/s in large sea areas) wave heights most likely did not exceed 10 m anywhere in the Baltic Sea (Soomere et al., 2008).*

*Therefore, I guess that wave heights between 11 and 12 m in Fig. 6 are completelyunrealistic for the February 2002 storm. It seems that the entire ensemble severely(by almost 2 m on average) overestimates wave heights in the northern Baltic*

[Figure]

*proper. Thus, I recommend to extensively comment this feature and to include a short insightinto measured or modelled wave heights in this area for storms of comparable prop-erties. Ideally, I would recommend to include a paragraph or two about extreme wave properties in the study area, following either (Tuomi et al., 2011) or (Björkqvist et al.,2018).*

Calibration and validation of WWIII driven by the UERRA/Harmonie-v1 forcing dataset against observations and the hindcast of the additional two storm events showed that the waves predicted with the Baltic Sea setup with UERRA/Harmonie-v1 and the unperturbed WRF-ARW hindcasts show reasonable wave heights. Our WWIII setup was calibrated with UERRA/Harmonie-v1 forcing, for this reason it was checked whether this calibration gives also reasonable results when driven with WRF-ARW. The wind in WRF-ARW is slightly stronger over land and near the coast, as it can be seen in Fig. 4. We adapted the roughness length over land according to the Corine land cover data set, but this gives only a small effect for the waves in the western part of the Baltic close to the land masses. The roughness length over the sea surface is assumed to be constant in the applied WRF-ARW setup. Under severe storm conditions, the roughness of the sea surface should increase in reality, resulting in a reduction of the wind speed due to higher momentum transfer. Reduced wind speeds limit the growth of the wave height. With a coupled WRF-WWIII setup, this effect could be taken into account, in our setup it is neglected. Perhaps this is one reason for the extreme wave heights in some representations. Based on one extreme event, it is also not possible to tune the perturbations of the model physics. This is why the WRF-ARW ensemble is potentially overdispersive, which we also mentioned in the manuscript. As can be seen in Figure 4, the wind speed over the Baltic proper in the extremest representations is above 28m/s. The time series shown in Figure 2 is at a different location. We will make this clearer in the manuscript and will discuss it more in detail that the extreme representations are potentially unrealistic. As the two additional storms were hindcasted with a 7km newer WRF-ARW version, which we will apply for our later application, a recalculation of the 2002 storm shows a maximum hs

of 9.5 m from an 11-member ensemble. The spread is larger than for the other two storm events which shows that the event is much more sensitive to perturbations.

*In particular, I recommend extending the message on page 5, line 5–6 towards a sound explanation that the model is essentially uncalibrated for the Baltic Sea conditions. This is mentioned in the last sentence before conclusions on page 14. The point of this sentence should be made very clear from Abstract to Conclusions. I stress that such a bias in the evaluated wave heights does not undermine the validity of most of the results but it should be made clear to the reader that single values of wave height (and even the ensemble average) do not necessarily match the wave properties in this storm.*
We will refer to the supplemental material and make it clearer that the WWIII setup was calibrated for the application with UERRA/Harmonie-v1, but a test with WRF-ARW wind also shows a reasonable performance. Concerning the extreme representation in some ensemble members, an additional discussion will be added about the uncalibrated ensemble spread in the WRF-ARW ensemble and about the fact that the effect of the roughness of the sea surface is not taken into account in the applied WRF-ARW setup.

*For the listed reasons I recommend moderate to major modifications to the manuscript. It is essential that the reader is informed (i) about some basic features of wave climate and extreme waves in the Baltic Sea and also (ii) that the simulations probably strongly overestimate wave heights and (iii) are performed specifically to study the spreading properties of ensembles, with no exact relevance to the actual wave heights during the simulation interval. An absolute must is to inquire the modelled data from a properly calibrated run (e.g., from the authors of Björkqvist et al., 2018) for the underlying location of Fig. 6 to give a minimum flavor of the possible bias.*
As proposed by both reviewers, additional information about the wave climate in the

[Figure]

Baltic Sea including citations to existing studies will be included into the introduction together with the mentioned points from the previous remarks about the potential overestimation of the spread in the atmospheric ensemble data resulting in potentially unrealistic wave heights in some members. We believe that a more detailed comparison to observations makes an inter-model comparison no longer a requirement.

*The text is written in fairly good English but reveals slight German accent in the form of very long sentences at places and missing of some articles in the text. It is mostly clear but still needs extensive polishing, especially closer to the end of the manuscript. As I am not native speaker, I only include a list of clear typos below.*
We will revise the text.

**2   Minor comments:**

*The paragraphs are at places very long. For example, the first paragraph of Introduction extends over 28 lines. It is recommended to split long paragraphs into shorter ones.*
The paragraphs will be splitted into shorter ones.

*The style of calendar days ("21. February 2002" on page 6, line 4 and "22nd to 24th of February" on the next line) should be unified.*
The style of the calendar days will be unified.

*The first two sentences of Abstract seem unnecessary*
The second sentence explains issues in wave modelling and is required for the third sentence which claims that we address these by the presented method. The first sentence shall put the second one into context. We believe this sort of introduction is

required to grasp the intention of the manuscript.

*Page 1, line 17: probably should be "and is described".*
Will be changed to "and is described".

*Line 23 and some other locations: some journals require comma after "e.g."*
Will be revised according to the requirement of the journal.

*Page 2, lines 32–34: the sentence does not make sense; possibly because of too strong German accent.*
Will be replaced by "' At the moment, the ensemble datasets in this project are limited in their temporal coverage or spatial resolution. It can be advantageous to be able to produce hindcasts of events whose spatiotemporal resolution is adapted to the requirements defined by a research objective."'

*Page 3, line 13: C3S has already been explained on page 2, line 22.*
Only the abbreviation will be used here.

*Line 20: probably full stops are not necessary in "21. February 2002" and similar expressions.*
Will be revised.

*Line 23 it is better to say that 0.36deg and 1deg denote the resolution of the relevant grid. Please do so also in several locations below where the size in degrees is given without any explanation.*
Will be adapted.

*Page 4, line 10: please specify the meaning of "writing 15 minutes output".*
Will be changed to: "and the model output interval is 15 minutes."

*Line 12: please explain what is meant under "the temporal impact" (probably the dependence of the solution on the time step).*
Will be changed to: "the dependence of the solution of the wave model on the temporal resolution of the wind data."

*Line 17: please specify the meaning of "Eta layers".*
Will be explained as a specific vertical coordinate system used for atmospheric models.

*Line 18–19: consider replacing the jargon-like expression "until fine scales develop" by a more explanative one. Please do so also in several occasions below to avoid clash in the meaning of, e.g., "finer scales are not represented" on page 6, line 9.*
We will replace "'scales'" by "'structures'" to avoid jargon.

*Page 5, line 1: to avoid misinterpretation, I suggest to mention that nesting of the wave model to the Baltic Sea is not really necessary for the hindcast of wave properties in the central and northern regions of this water body because very little wave energy penetrates through the Danish straits.*
The wave model output will be used as input for a model of the entire Baltic Sea which also covers a part north of the Danish straits. In this region, we also want to have reasonable wave parameters. This will be made clear in the manuscript.

*The reasoning on lines 2–6 is only partially relevant for the conditions of the Baltic*

*Sea.*
It explains the procedure used for setting up the wave model. ERA5 is a global reanalysis and the procedure could be applied also to other regions in the world.

*Line 7: while most of the model setup is obviously fine for the Baltic Sea, please com-ment on the adequacy of the use of the chosen frequency range for this water body. Wave modellers usually substantially extend the frequency space here. The team of the Finnish Meteorological Institute normally uses 35 frequencies (Laura Tuomi et al.,many papers) and some research in subbasins of the Baltic Sea even 42 frequencies (0.0418–2.08 Hz, Soomere, 2005). It is probably not necessary to cover such an extended range. However, insufficient coverage of short waves may lead to too slow wave growth under rapidly increasing wind conditions.*
The discretization with 42 frequencies (0.0418–2.08 Hz, Soomere, 2005) together with a finer resolution of the directions (36 every 10deg) was tested. In the supplemental material, the outcome is visible. It brings additional 10cm in the significant wave height for the Rafael storm, which is underestimated by about 90cm with the UERRA/Harmonie-v1 wind. The shortcoming of this finer discretization is a prolongation of the calculation time, which was 4 times of the one with the ERA5 equivalent discretization. For computational reasons, we used the ERA5 discretization.

*Lines 14–16: the message of the entire sentence is technically clear but seems misplaced or even irrelevant.*
Will be revised.

*Page 7, line 7: "these".*
Adapted.

*Page 10, lines 19–20, the sentence "Compared to ERA5, the overall spatial pattern is comparable" does not make sense to me.*
Will be replaced by "'The overall spatial pattern of the significant wave height is comparable between ERA5 and the WRF ensemble members.'"

*Page 11, lines 2–5: the reasoning is almost irrelevant for the Baltic Sea conditions andshould be left out. Instead, it should be emphasized that strong swells are infrequentin the Baltic Sea (see, e.g., Broman et al., 2006; Soomere et al., 2012) and thus deviations in the hindcast or forecast driven by the accuracy of the representation of swells are usually not very large in this water body.*
We see the manuscript as a demonstration for the procedure to produce ensemble hindcasts. ERA5 is global and the procedure is applicable in general worldwide. This is why we also have to mention potential shortcomings if applying the procedure to other regions. We will add a subsentence "', which should, however, be more relevant for different regions of interest where swell plays a larger role.'"

*Page 12, line 14: something is wrong with "500 choose N possibilities exist".*
This is an expression from stochastics, we will replace it by the mathematical notation $\binom{500}{N}$ to avoid confusion.

*Page 13, line 10–12: the sentence is unclear.*
Will be revised.

*Line 13: "developed"; also, the entire sentence remains partially unclear starting from "why".*
Will be revised.

*Lines 16–17: the concluding sentence of the subsection should be made clearer.*
Will be changed to "'Depending on the application, the ensemble size needs to be selected by a compromise between the robustness of the uncertainty estimate and the computational cost.'"

*Line 18: use "on" instead of "onto".*
Will be changed.

*Page 14, line 7: please specify what is meant under "The higher temporal resolutions do not differ so much." Also, the subsequent sentences contain too much jargon.*
Will be revised.

*Line 14: "orography of the coastlines" sounds weird as the height of the coastline is just zero; also: use "Baltic Sea".*
Will be revised.

*Line 15: spatial pattern of what?*
of the significant wave height.

*Line 5 or another appropriate place: please stress that an uncalibrated (for the Baltic Sea conditions) wave model was used but still the results about the spread are valid.*
We will refer to the supplemental material and the issue with the spread will be discussed there.

*Page 16, line 1: remove "by this fact".*
Removed.
*Lines 1–2: the message of the sentence "As the first twelve hours are not used, be-cause of the model spin-up, this is not really a shortcoming." remains unclear.*
Will be revised.

*Line 14: correct "atmopsheric".*
Corrected.

*Line 20: correct "possbile".*
Corrected.

Please also note the supplement to this comment:
https://www.ocean-sci-discuss.net/os-2019-76/os-2019-76-AC2-supplement.pdf

**Supplement:**

**1    Supplemental material**

This supplement describes the WWIII setup for the Baltic Sea applied in the presented study. Two different versions of WWIII were used: While v6.07 was applied for the figures in the article and the model validation shown here, the model calibration was performed by version 5.16 and the parameters were not readjusted after switching to the new version. We used the source term packages defined after the switch file Ifremer2, which is part of the model source code. This includes wind input and dissipation (ST4) after Ardhuin et al. (2010) and the SHOWEX bottom friction scheme (BT4) after Ardhuin et al. (2003). For the latter, a sediment map with a median grain size (D50) based on EMODnet data (Fig. 1) is applied to define a spatially dependent bottom friction. Wind data from the UERRA/Harmonie-v1 analysis/forecasts (Ridal et al., 2017) were used. Analyses are available every six hours (00,06,12,18UTC) and were combined with the forecasts for +1 to +5 hours to create a dataset of hourly data. Model calibration and validation is based on this dataset.

[Figure]

**Figure 1.** Grain size in Krumbein Phi scale used for the SHOWEX bottom friction scheme.

**1.1    Calibration**

Wind input and dissipation from the ST4 physics were tuned specifically to the Baltic Sea region. A sensitivity study of the impact of the wind input parameter betamax and the dissipation parameter SDSC2 has been conducted for this purpose, comparing to observed significant wave heights for January 2017, which were available from the CMEMS data base. The bias, rmse, correlation and the scatter index are used as objective scores for model validation acording to Zambresky (1989). The parameters were varied inside their valid range (Gorman and Oliver, 2018). The figures 3, 4 and 5 show the scores over the valid range of the betamax parameter. If increasing betamax, an improvement of the scores is visible, but for the eastern of the three stations betamax should be much higher than for the western stations. An overall good performance for the entire Baltic Sea is the aim of the calibration, what can not be achieved by only tuning betamax.   The sensitivity study shows that a compromise between the wind input parameter and the dissipation parameter has to be found. Figure 6 shows the sensitivity in the scores for two different betamax values over a wide range of the dissipation parameter SDSC2. Additionally, sensitivity tests for the

**CMEMS stations with significant wave height**

[Figure]

**Figure 2.** Station data available from CMEMS.

[Figure]

**Figure 3.** Verification scores for January 2017 for sensitivity study of wind input parameter betamax for station Bothnian Sea.

sheltering parameter taushelter and the tail factor (FXFM3) have been done. Based on tests with different parameter sets and a subjective interpretation, the four parameters were changed with regard to the default parameter set (Table 1).

As the calibration of the model is based on the previous model version 5.16, January 2017 has been recalculated with the actual version 6.07. The final choice of the parameter set is based on a subjective interpretation of the different test parameter sets.

[Figure]

**Figure 4.** Verification scores for January 2017 for sensitivity study of wind input parameter betamax for station Northern Baltic.

[Figure]

**Figure 5.** Verification scores for January 2017 for sensitivity study of wind input parameter betamax for station Vahemadal.

An objective calibration procedure like the one demonstrated by Gorman and Oliver (2018) was out of the scope of this study. For our purposes, the wave model shows a satisfactory model performance based on the demonstrated calibration procedure. To test if the calibration of the WWIII model based on the UERRA/Harmonie-v1 wind data is also appropriate for the wind fields hindcasted with the WRF-ARW model, one month of data were produced. For this purpose, every six hours, an eighteen

[Figure]

**Figure 6.** Verification scores for January 2017 for sensitivity study of dissipation parameter SDSC2 for station Bothnian Sea.

**Table 1.** Adapted parameter sets of the WAVEWATCH III® setup with Ifremer2 source term packages for the Baltic Sea based on the sensitivity study.

| Parameter | betamax | SDSC2 | taushelter | FXFM3 |
|---|---|---|---|---|
| Default value | 1.43 | 2.2E-5 | 0.3 | 2.5 |
| Adjusted value | 1.60 | 1.8E-5 | 0.2 | 5 |

hour hindcast was produced for January 2019. From these 18h hindcasts, the first 12h were considered as a model spinup and the last six hours were taken and combined to an hourly dataset.

The performance of WWIII with the WRF-ARW wind is also satisfactory based on the UERRA/Harmonie-v1 calibration. Over land areas, the wind is stronger in the WRF-ARW setup as in UERRA/Harmonie-v1. A comparison with the Corine land cover
5 dataset showed that the roughness length for different land usages are lower in the WRF-ARW setup, which can explain the higher wind speeds over land. A test with modified roughness lengths showed only a small impact in the western part of the Baltic Sea, as this region is an enclosed coastal-ocean area with close proximity of the sea grid cells to land. The roughness of the sea surface is assumed to be constant in the applied WRF-ARW setup. Under strong storm conditions, the roughness of the sea surface should increase, which has an effect on the wind speed. This could be one reason for the outlying members
10 in the demonstrated example of the significant wave height in the 2002 storm surge event. Nevertheless, the wave height in WWIII with UERRA/Harmonie-v1 is very close to the one with the WRF-ARW setup for the presented 2002 event. A coupled WWIII/WRF-ARW model would include the effect of the sea surface on its roughness length.

[Figure]

**Figure 7.** Observation, WWIII with default parameters driven with WRF-ARW (0.063°) wind, WWIII calibration for UERRA/Harmonie-v1 with WRF-ARW wind and UERRA/Harmonie-v1 wind for January 2019 at station Northern Baltic.

**1.2 Hindcasts of Storms Rafael and Toini**

For the 2002 storm surge event we presented, observations for wave parameters are not available. As proposed by the reviewers, two additional events were hindcasted. Ensemble generation procedure number five presented in the article was repeated here for hindcasting the storms Rafael and Toini, which provoked the highest observed significant wave heights in the Baltic proper.

5  Detailed information about these two storm events can be found in Björkqvist et al. (2017). The newest WRF version v4.1.1 was used with 7km resolution. An obstruction grid based on the GSHHS shoreline dataset and generated with the gridgen[1] software is applied to take unresolved orography into account in the WWIII wave model. The WRF-ARW model runs were started for the Rafael storm on 21 December 2004 00UTC and finished on 24 December 2004 00UTC. SST is updated over the integration time, whereas in the article, the SST was kept constant after initialisation. Ten members from the ERA5 Ensemble

10  of Data Assimilation were used as initial conditions for the perturbed runs together with stochastic perturbations. In addition, the entire January 2004 was hindcasted without the ensemble approach, using the UERRA/Harmonie-v1 wind data. Initial conditions in the wave model for the Rafael storm hindcast were taken from this previous model run on 21 December 2004 12UTC. Runs with WRF-ARW unperturbed and perturbed 5-minutes wind forcings were done for the period 21 December 2004 12UTC until 24 December 2004 00UTC. Sea ice data were taken from the ERA5 reanalysis. The same model setup was

15  applied for the Toini storm. WRF-ARW runs started on 10 January 2017 00UTC and ended on 13 January 2017 00UTC. WWIII was driven with these data for the period 10 January 2017 12UTC until 13 January 2017 and initial conditions for WWIII were taken from a previous run for the entire January 2017 driven by UERRA/Harmonie-v1 winds. The hindcast of the storm Rafael shows an overall good performance with an underestimation of the peak wave height with UERRA/Harmonie-v1 by about 90 cm and by WRF-ARW of about 80 cm. One ensemble member matches the observed maximum. A finer discretization

20  of the energy spectrum as proposed by Soomere (2005) increases the maximum wave height with the UERRA/Harmonie-v1 forcing by about 10 cm. The computation with this finer discretization took nearly four times as long as with the ERA5-equivalent discretization. For both wind data sets, the same parameter set is used. The strongest member slightly exceeds the observed significant wave height. Much higher waves in single ensemble members, as they occurred for the 2002 event, are not
* * *
[1]https://github.com/NOAA-EMC/gridgen

[Figure]

**Figure 8.** Significant wave height [m] at Arkona and Northern Baltic stations for December 2004.

[Figure]

**Figure 9.** Significant wave height [m] at Arkona and Northern Baltic stations for Storm Rafael driven with UERRA/Harmonie-v1 wind and WRF-ARW ensemble hindcasts with eleven perturbed and one unperturbed runs.

generated in this ensemble hindcast. As the ensemble size is very small, it cannot be excluded that realisations with larger wave heights are possible. For the Toini storm, the unperturbed run generates too small wave heights. Nearly all perturbed members produce larger wave heights. The observed wave height is well covered by the ensemble. When we hindcasted the 2002 event with this 7 km setup with 11 perturbed members, one realisation showed a maximum of 9.5 m. This suggests that the 2002

5  event reacts more sensitive to the perturbations than the two other storm events.

[Figure]

**Figure 10.** Significant wave height [m] at the Northern Baltic for Storm January 2017 and for storm Tioni driven with UERRA/Harmonie-v1 wind and WRF-ARW ensemble hindcasts with eleven perturbed runs and one unperturbed one.

**References**

Ardhuin, F., H C Herbers, T., O'Reilly, W., and Jessen, P.: Swell Transformation across the Continental Shelf. Part I: Attenuation and Directional Broadening, Journal of Physical Oceanography, 33, 1921, https://doi.org/10.1175/1520-0485(2003)033<1921:STATCS>2.0.CO;2, 2003.

Ardhuin, F., Rogers, E., Babanin, A. V., Filipot, J.-F., Magne, R., Roland, A., van der Westhuysen, A., Queffeulou, P., Lefevre, J.-M., Aouf, L., and Collard, F.: Semiempirical Dissipation Source Functions for Ocean Waves. Part I: Definition, Calibration, and Validation, Journal of Physical Oceanography, 40, 1917–1941, https://doi.org/10.1175/2010JPO4324.1, https://doi.org/10.1175/2010JPO4324.1, 2010.

Björkqvist, J.-V., Tuomi, L., Tollman, N., Kangas, A., Pettersson, H., Marjamaa, R., Jokinen, H., and Fortelius, C.: Brief communication: Characteristic properties of extreme wave events observed in the northern Baltic Proper, Baltic Sea, Natural Hazards and Earth System Sciences, 17, 1653–1658, https://doi.org/10.5194/nhess-17-1653-2017, https://www.nat-hazards-earth-syst-sci.net/17/1653/2017/, 2017.

Gorman, R. M. and Oliver, H. J.: Automated model optimisation using the Cylc workflow engine (Cyclops v1.0), Geoscientific Model Development, 11, 2153–2173, https://doi.org/10.5194/gmd-11-2153-2018, https://www.geosci-model-dev.net/11/2153/2018/, 2018.

Ridal, M., Olsson, E., Unden, P., Zimmermann, K., and Ohlsson, A.: Uncertainties in Ensembles of Regional Re-Analyses - Deliverable D2.7 HARMONIE reanalysis report of results and dataset, http://www.uerra.eu/component/dpattachments/?task=attachment.download&id=296, 2017.

Soomere, T.: Wind wave statistics in Tallinn Bay, Boreal Environment Research, 10, 103–118, 2005.

Zambresky, L.: A verification study of the global WAM model December 1987 - November 1988, p. 86, https://www.ecmwf.int/node/13201, 1989.

---

## Author Response (AR2)

Dear Referee #2, Dear editor Dr. Wolf,

Thank you for your comments. Please find below our answers in *italic.*

The manuscript has considerably improved. The revised version includes reflections to most of my comments to the first version as well as a number of valuable adjustments and amendments according to the recommendations of the other referee. There are still a few concerns and several mostly minor issues that need adjustment before the manuscript can be accepted. It is said both in Abstract and Conclusions that the event addressed in the manuscript provoked a severe storm surge in February 2002. Please insert the relevant information (how high surge and where was observed) or just say it was a relatively strong storm event. For example, nothing specific happened in Latvian waters.
*In the text it is now said that 2002 was a "relatively strong storm event".*

Modelled wave heights above 11 m in some ensemble members: I agree with the explanation of the authors that (i) some realisations of wind fields contained wind speeds of 28 m/s and such winds may drive very high waves in the Baltic Sea, (ii) the use of constant roughness length over the sea surface in the wave model could unrealistically contribute to the wave energy growth, and (iii) as a result, the spread of the ensemble may be fairly large (e.g., page 11, lines 20-23). However, point (ii) in plain English means that the used model setup most likely systematically overestimates wave heights in some storms, including the analysed event. I strongly recommend to say that explicitly both in the body text and in conclusions.
*In the supplemental material, we have shown that the wave model driven with both datasets, UERRA/Harmonie-v1 and WRF, shows a relatively good performance, and we hindcasted additionally two storm events which provoked the highest ever measured significant wave height in the Baltic Sea. Both of these storm events are well represented in the ensemble hindcasts. We agree that the constant roughness length can provoke systematical overestimations in some storm events. For the presented event, we used WRF as well as UERRA/Harmonie-v1. The latter dataset provoked even a slightly higher significant wave height than the unperturbed WRF run. The significant wave heights are at least for the deterministic runs with WRF and UERRA/Harmonie-v1 in a reasonable order close to the two most severe observed events. Of course, with a different dynamical evolution of the storm, we think that some of the ensemble members can be stronger affected by the negligence of the changing surface roughness than others. We added as recommended in the body text as well as in the conclusions a phrase mentioning that the constant roughness length can lead to systematical overestimations in some storms. This is a shortcoming in the actual setup but can be improved by coupling the atmospheric with the wave model, as also mentioned in the manuscript.*

Lines 4-7 of Abstract are too unspecific. Consider saying, for example: In this study, we apply an atmospheric downscaling to (i) add regional details to the wind field, (ii) increase the temporal resolution of the wind fields, (iii) provide a more detailed representation of transient events such as storms, and (iv) generate ensembles with perturbed atmospheric conditions which allow for a spatiotemporally variable uncertainty estimation. Also please insert the explanation of WRF already in Abstract.
*We changed the phrase as proposed and added the explanation of the WRF abbreviation to the abstract.*

Consider saying in the Baltic Sea in the title
*We see the Baltic Sea as well as the application of the wave model more than an example. ERA5 is a global reanalysis. The approach can be applied to other world regions. An ensemble approach is also interesting for other variables, e.g. for storm surge hindcasts by driving a hydrodynamical model. We think that the method could be of interest for a wider audience, why we prefer to not mention Baltic Sea in the title.*

The style of using -isa/iza- etc. should be unified. For example, there is realisations on page 1, line 19 but parameterizations on page 2, line 17.
*-iza is now used everywhere.*

Page 3, line 25 and in several occasions below: as the length of a degree along latitudes and along longitudes differs by a factor of 2 in the Baltic Sea region, please make sure to the readers that the grid cells in the used models have quite large aspect ratio, the length of their sides in N-S direction roughly twice as long as in the E-W directions. Also, consider using the degree sign instead of deg in the entire manuscript.
*A footnote was added concerning the aspect ratio of the grid cells in this region. The ° sign is used instead of deg.*

Page 3, line 29: probably should be and interpolated
*Interpolated data were extracted from the database.*

Page 4, line 27: use Greek $\eta$
*In the literature, this model is written "ETA model", e.g. in the WRF user guide. This is why we prefer to keep it in this way.*

Page 5, line 11: must be unrealistic
*Corrected*

Page 7, lines 13-15: the sentence is unclear, in particular the expression why some directions.
*Changed to: From ERA5, also the wind speed from the closest grid cell of the 0.25° grid is plotted. The initial conditions were prepared with the WRF preprocessing system, are visible and can have slightly different values from taking simply the closest grid cell.*

Page 7, lines 18-19: The LAF ensemble not only shows a very small spread. It is obviously underdispersive (and saying that an indication that these ensembles could be underdispersive is probably not sufficient). And this is by no means good from the viewpoint of forecast or hindcast.
*Sentence changed to: Compared to the ERA5 EDA members, it demonstrates that these ensembles are under-dispersive. The uncertainty is underestimated by applying these approaches.*

Page 7, line 20: Here and in several occasions below (page 11, line 13; page 11, line 16; page 17, lines 4, 7, and possibly in some other locations) the use of the word comparable is not appropriate. It is possible to compare quantities that differ by tens and hundreds of times. Please be more specific and use, whenever applicable, quantitative comparisons.
*Changed for the mentioned locations.*

Page 8, lines 14-15: I do not recognise the point of the sentence Comparing a ten with a thirty member ensemble is not really a fair comparison.
*In the paragraph about the robustness of the ensemble spread, it is shown that the quality of the representation of the uncertainty depends on the ensemble size. Sentence adapted to: "Comparing a ten with a thirty member ensemble is not really a fair comparison, as a too small ensemble size leads to an undersampling of the uncertainty."*

Page 8, line 15: delete number; approach 6 is enough
*Number deleted.*

Page 9, line 1: delete of all ensemble members
*of all ensemble members deleted.*

Page 9, lines 3 and 4: delete number; approach 6 is enough
*Number deleted.*

Page 9, line 9: replace station by location
*replaced*

Page 11, line 2: probably variability in (wind and) wave properties is meant
*The paragraph is about the wave fields. Changed to: Especially the maximum of the (wind speed and the) significant wave height varies strongly between the different ensemble realisations.*

Page 11, line 4: probably position of wave height maxima or similar is meant
*Changed to: Differences in the wave fields of the ensemble members can be due to a different dynamical evolution of the storm or due to different tracks in the atmospheric model members*

Page 11, line 8: is in this region with does not make sense to me; probably represent is meant
*Adapted in the proposed way.*

Page 12, line 1: remove the fact
*Removed*

Page 13, line 10: remove the second apply
*Removed*

Page 14, line 6: please give an extended formula for the expression (500//N); I guess that most of the potential readers do not associate it with math/combinatorics.
*Changed to: The number of possible combinations of selecting N out of 500 members can be determined by using the binomial coefficient 500... .*

Page 14, line 13: should be a higher
*Adapted*

Page 15, line 4: remove numerical, replace can be counted in seconds by is less than a minute and say few minutes at the end of the line
*Adapted, but numerical was kept to avoid confusion with the time step for writing the model output.*

Page 15, line 8: explain SST
*Sea-surface temperature added*

Page 15, line 10: should be an erroneous
*Corrected*

Page 16, last sentence to the caption of Fig. 10 could be simply The northern part of the Gulf of Bothnia is covered by sea ice
*Changed as proposed*

Page 16, line 4: better say high resolution setup
*Changed as proposed*

Page 16, line 5: better remove what explains the higher but also the lower wave heights and say: Wave height maxima
*Changed as proposed*

Page 17, line 16: consider saying than the one obtained using
*Changed as proposed*

[revised manuscript text omitted]

indicates that the choice of the 15 minutes resolution is a reasonable compromise between a good representation of the extreme values and file size.

A stronger impact can be expected from the spatial resolution of the driving wind fields, because a coarser resolution of the atmospheric model can be assumed to produce lower extreme wind speeds as a grid cell represents the average value over the area it covers. By adapting, for example, the parameter betamax which describes the maximum value of wind-wave coupling, this difference can be compensated for. A better representation of the complex coastline of the Baltic sea as well as of the various islands is given by the higher resolved WRF model. For this reason, a difference in the spatial pattern of the significant wave height can be expected. A test with the coarsest (0.252°̰) and the highest resolution (0.063°̰) produced in this study has been conducted. The same parameter sets were used, as a calibration is not possible based on the short period hindcasted with WRF. Figure 10 shows the difference between these two forcings on one timestep in the significant wave height. One grid cell of the coarser WRF setup contains 16 grid cells of the  h̲i̲g̲h̲ ̲r̲e̲s̲o̲l̲u̲t̲i̲o̲n̲ ̲s̲e̲t̲u̲p̲.̲ ̲W̲a̲v̲e̲ ̲h̲e̲i̲g̲h̲t̲ m̲a̲x̲i̲m̲a̲ 
[revised manuscript text omitted]